# MaestroMotif: Skill Design from Artificial Intelligence Feedback

**Martin Klissarov**[1,5]**, Mikael Henaff**[2]**, Roberta Raileanu**[2]**,**
**Shagun Sodhani**[2]**, Pascal Vincent**[1,2]**, Amy Zhang**[2,3]**, Pierre-Luc Bacon**[1,4]**,**
**Doina Precup**[1,5,8]**, Marlos C. Machado**[*,6,7,8]**, Pierluca D'Oro**[*,1,2,4]

[1] Mila, [2] Meta, [3] University of Texas Austin, [4] Université de Montréal,
[5] McGill University, [6] University of Alberta, [7] Amii, [8] Canada CIFAR AI Chair

## Abstract

Describing skills in natural language has the potential to provide an accessible way to inject human knowledge about decision-making into an AI system. We present MaestroMotif, a method for AI-assisted skill design, which yields high-performing and adaptable agents. MaestroMotif leverages the capabilities of Large Language Models (LLMs) to effectively create and reuse skills. It first uses an LLM's feedback to automatically design rewards corresponding to each skill, starting from their natural language description. Then, it employs an LLM's code generation abilities, together with reinforcement learning, for training the skills and combining them to implement complex behaviors specified in language. We evaluate MaestroMotif using a suite of complex tasks in the NetHack Learning Environment (NLE), demonstrating that it surpasses existing approaches in both performance and usability.

## 1 Introduction

Bob wants to understand how to become a versatile AI researcher. He asks his friend Alice, a respected AI scientist, for advice. To become a versatile AI researcher, she says, one needs to practice the following skills: creating mathematical derivations, writing effective code, running and monitoring experiments, writing scientific papers, and giving talks. Alice believes that, once these different skills are mastered, they can be easily combined following the needs of any research project.

Alice is framing her language description of how to be a versatile researcher as the description of a set of skills. This often happens among people, since this type of description is a convenient way to exchange information on how to become proficient in a given domain. Alice could instead have suggested what piece of code or equation to write, or, at an even lower level of abstraction, which keys to press; but she prefers not to do it, because it would be inconvenient, time-consuming, and likely tied to specific circumstances for it to be useful to Bob. Instead, describing important skills is easy but effective, transmitting large amounts of high-level information about a domain without dealing with its lowest-level intricacies. Understanding how to do the same with AI systems is still a largely unsolved problem.

Recent work has shown that systems based on Large Language Models (LLMs) can combine sets of skills to achieve complex goals (Ahn et al., 2022; Wang et al., 2024). This leverages the versatility of LLMs to solve tasks *zero-shot*, after the problem has been *lifted* from

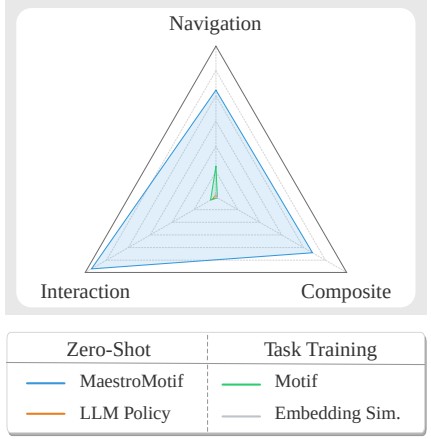

Figure 1: Performance across NLE task categories. MaestroMotif largely outperforms existing methods zero-shot, including the ones trained on each task.

---

*Equal supervision. Correspondence to: `martin.klissarov@mail.mcgill.ca`.

the low-level control space, which they have difficulty handling, to a high-level skill space grounded in language, to which they are naturally suited. However, humans cannot communicate skills to these systems as naturally as Alice did with Bob. Instead, such systems typically require humans to solve, by themselves, the *skill design problem*, the one of crafting policies subsequently used by the LLM. Designing those skills typically entails very active involvement from a human, including collecting skill-specific data, developing heuristics, or manually handling reward engineering (Ahn et al., 2022). Thus, existing frameworks for designing low-level skills controlled by LLMs require technical knowledge and significant amounts of labor from specialized humans. This effectively reduces their applicability and generality.

In this paper, we introduce the paradigm of *AI-Assisted Skill Design*. In this paradigm, skills are created in a process of human-AI collaboration, in which a human provides a natural language description of the skills and an AI assistant automatically converts those descriptions into usable low-level policies. This strategy fully leverages the advantages of both human-based and AI-based skill design workflows: it allows humans to inject important prior knowledge about a task, which may enhance safety and performance for the resulting agents even in the absence of optimal AI assistants; at the same time, it automates the lowest-level and more time-consuming aspects of skill design.

Based on this paradigm, we propose MaestroMotif, a method that uses LLMs and reinforcement learning (RL) to build and combine skills for an agent to behave as specified in natural language. MaestroMotif uses an LLM's feedback to convert high-level descriptions into skill-specific reward functions, via the recently-proposed Motif approach (Klissarov et al., 2024). It then crafts the skills by writing Python code using an LLM: first, it generates functions for the initiation and termination of each skill; then, it codes a policy over skills which is used to combine them. During RL training, the policy of each skill is optimized to maximize its corresponding reward function by interacting with the environment. At deployment time, MaestroMotif further leverages code generation via an LLM to create a policy over skills that can combine them almost instantaneously to produce behavior, in zero-shot fashion, as prescribed by a human in natural language.

MaestroMotif thus takes advantage of RL from AI feedback to lift the problem of producing policies from low-level action spaces to high-level skill spaces, in a significantly more automated way than previous work. In the skill space, planning becomes much easier, to the point of being easily handled zero-shot by an LLM that generates code policies. These policies can use features of a programming language to express sophisticated behaviors that could be hard to learn using neural networks. In essence, MaestroMotif crafts and combines skills, similarly to motifs in a composition, to solve complex tasks.

We evaluate MaestroMotif on a suite of tasks in the Nethack Learning Environment (NLE) (Küttler et al., 2020), created to test the ability to solve complex tasks in the early phase of the game. We show that MaestroMotif is a powerful and usable system: it can, without any further training, succeed in complex navigation, interaction and composite tasks, where even approaches trained for these tasks struggle. We demonstrate that these behaviors cannot be achieved by baselines that maximize the game score, and we perform an empirical investigation of different components our method.

## 2 BACKGROUND

A language-conditioned Markov Decision Process (MDP) (Liu et al., 2022) is a tuple $\mathcal{M} = (\mathcal{S}, \mathcal{A}, \mathcal{G}, r, p, \gamma, \mu)$, where $\mathcal{S}$ is the state space, $\mathcal{A}$ is the action space, $\mathcal{G}$ is the space of natural language task specifications, $r : \mathcal{S} \times \mathcal{G} \to \mathbb{R}$ is the reward function, $p : \mathcal{S} \times \mathcal{A} \to \Delta(\mathcal{S})$ is the transition function, $\gamma \in (0, 1]$ is the discount factor, $\mu \in \Delta(\mathcal{S})$ is the initial state distribution.

A skill can be formalized through the concept of option (Sutton et al., 1999; Precup, 2000). A deterministic Markovian option $\omega \in \Omega$ is a triple $(\mathcal{I}_\omega, \pi_\omega, \beta_\omega)$, where $\mathcal{I}_\omega : \mathcal{S} \to \{0, 1\}$ is the initiation function, determining whether the option can be initiated or not, $\pi_\omega : \mathcal{S} \to \Delta(\mathcal{A})$ is the intra-option policy, and $\beta_\omega : \mathcal{S} \to \{0, 1\}$ is the termination function, determining whether the option should terminate or not. Under this mathematical framework, the skill design problem is equivalent to constructing a set of options $\Omega$ that can be used by an agent. The goal of the agent is to provide a policy over options $\pi : \mathcal{G} \times \mathcal{S} \to \Omega$. Whenever the termination condition of an option is reached, $\pi$ selects the next option to be executed, conditioned on the current state. The performance of such a policy is defined by its expected return $J(\pi) = \mathbb{E}_{\mu, \pi, \Omega}[\sum_{t=0}^{\infty} \gamma^t r(s_t)]$.

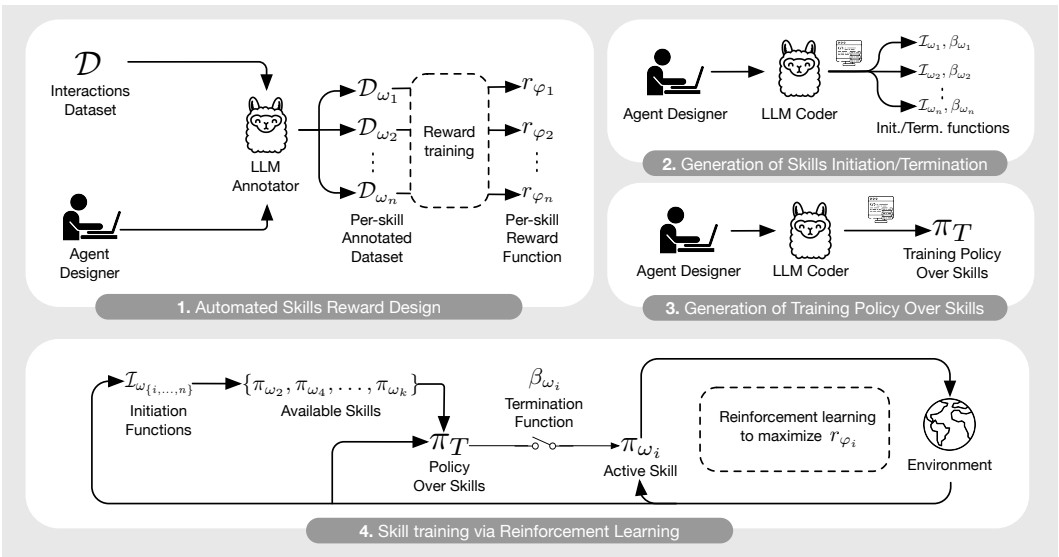

Figure 2: AI-assisted Skill Design with MaestroMotif. **1.** An agent designer provides skills descriptions, which get converted to reward functions $r_{\varphi_1}$ by training on the preferences of an LLM on a dataset of interactions. **2.** The agent designer describes initiation and termination functions, $\mathcal{I}_{\omega_{\{1,\dots,n\}}}$ and $\beta_{\omega_{\{1,\dots,n\}}}$ to the LLM, which instantiates them by generating code. **3.** The agent designer describes a train-time policy over skills $\pi_T$ which the LLM generates via coding. **4.** Each skill policy $\pi_{\omega_i}$ is trained to maximize its corresponding reward $r_{\varphi_i}$. Whenever a skill terminates (see open/closed circuit), the policy over skills chooses a new one from the set of available skills.

In the AI-Assisted Skill Design paradigm, an agent designer provides a set of natural language prompts $\mathcal{X} = \{x_1, x_2, \dots, x_n\}$. Each prompt consists of a high-level description of a skill. An AI system should implement a transformation $f : \mathcal{X} \to \Omega$ to convert each prompt into an option. Note that the ideas and method presented in this paper generalize to the partially-observable setting and, in our experiments, we learn memory-conditioned policies.

## 3 METHOD

MaestroMotif leverages AI-assisted skill design to perform zero-shot control, guided by natural language prompts. To the best of our knowledge, it is the first method that, while only using language specifications and unannotated data, is able to solve end-to-end complex tasks specified in language. Indeed, RL methods trained from scratch cannot typically handle tasks specified in language (Touati et al., 2023), while LLM-based methods typically feature labor-intensive methodologies for learning low-level control components (Ahn et al., 2022; Wang et al., 2024). MaestroMotif combines the capability of RL from an LLM's feedback to train skills with an LLM's code generation ability which allows it to compose them at will. We first introduce MaestroMotif as a general method, describing its use for AI-assisted skill design and zero-shot control, then discussing its implementation.

### 3.1 AI-ASSISTED SKILL DESIGN WITH MAESTROMOTIF

MaestroMotif performs AI-assisted skill design in four phases shown in Figure 2. It leverages LLMs in two ways: first to generate preferences, then to generate code for initiation/termination functions and for a training-time policy over skills. It then uses these components to train skills via RL.

**Automated Skills Reward Design** In the first phase, an agent designer provides a description for each skill, based on their domain knowledge. Then, MaestroMotif employs Motif (Klissarov et al., 2024) to create reward functions specifying desired behaviors for each skill: it elicits preferences of an LLM on pairs of interactions sampled from a dataset $\mathcal{D}$, forming for each skill a dataset of skill-related preferences $\mathcal{D}_{\omega_i}$, and distilling those preferences into a skill-specific reward function $r_{\varphi_i}$

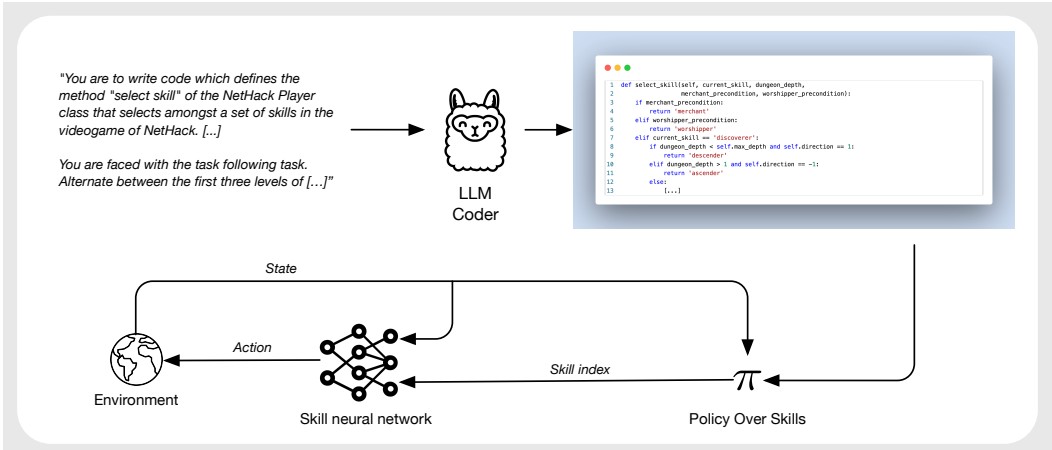

Figure 3: Generation of policy over skills during deployment. The LLM takes a task description and a template as an input, and implements the code for the policy over skills as a skill selection function. Running the code yields a policy over skills that commands a skill neural network by sending the appropriate skill index. Initiation and termination functions, determining which skills can be activated and when a skill execution should terminate, are omitted from the diagram.

by minimizing the negative log-likelihood, i.e., using the Bradley-Terry model:

$$\mathcal{L}(\varphi_i) = -\mathbb{E}_{(s_1,s_2,y)\sim\mathcal{D}_{\omega_i}}\left[\mathbb{1}[y=1]\log P_{\varphi_i}[s_1 \succ s_2] + \mathbb{1}[y=2]\log P_{\varphi_i}[s_2 \succ s_1]\right], \quad (1)$$

where $y$ is an annotation generated by an LLM annotator and $P_\varphi[s_a \succ s_b] = \frac{e^{r_{\varphi_i}(s_a)}}{e^{r_{\varphi_i}(s_a)}+e^{r_{\varphi_i}(s_b)}}$ is the estimated probability of preferring a state to another (Bradley & Terry, 1952).

**Generation of Skill Initiation/Termination** While a reward function can steer the behavior of a skill when it is active, it does not prescribe when the skill can be activated or when it should be terminated. In the options framework, this information is provided by the skill initiation and termination functions. MaestroMotif uses an LLM to transform a high-level specification into code that defines the initiation function $\mathcal{I}_{\omega_i}$ and termination function $\beta_{\omega_i}$ for each skill.

**Generation of training-time policy over skills** To be able to train skills, MaestroMotif needs a way to decide which skill to activate at which moment. While skills could be trained in isolation, having an appropriate policy allows one to learn skills using a state distribution closer to what will be needed during deployment, and to avoid redundancies. For instance, suppose the agent designer decided to have a two-skill decomposition, such that skill A's goal can only be achieved after skill B's goal is achieved; if they are not trained together, skill A would need to learn to achieve also the goal of skill B, nullifying any benefits from the decomposition. To avoid this, MaestroMotif leverages the domain knowledge of an agent designer, which gives a language specification of how to interleave skills for them to be learned more easily. From this specification, MaestroMotif crafts a policy over skills to be used at training time, $\pi_T$, which, as with the previous phase, is generated as code by an LLM.

**Skills training via RL** In the last phase of AI-assisted skill design, the elements generated in the previous phases are combined to train the skill policies via RL. Following the call-and-return paradigm (Sutton et al., 1999), the training policy $\pi_T$ decides which skill to execute among the ones deemed as available by the initiation functions $\mathcal{I}_{\omega_{\{1,\ldots,n\}}}$. Then, the skill policy $\pi_{\omega_i}$ of the selected skill gets executed in the environment and trained to maximize its corresponding reward function $r_{\varphi_i}$ until its termination function $\beta_{\omega_i}$ deactivates it. Initialized randomly at the beginning of the process, each skill policy will end up approximating the behaviors originally specified in natural language.

### 3.2 Zero-shot Control with MaestroMotif

After AI-assisted skill design, MaestroMotif has generated a set of skills, available to be combined. During deployment, a user can specify a task in natural language; MaestroMotif processes this

language specification with a code-generating LLM to produce and run a *policy over skills* $\pi$ that, without any additional training, can perform the particular task.

The policy over skills $\pi$ is then used, together with the skill policies $\pi_{\omega_{\{i,...,n\}}}$, initiation functions $\mathcal{I}_{\omega_{\{i,...,n\}}}$, and termination functions $\beta_{\omega_{\{i,...,n\}}}$, built through AI-assisted skill design, to compose the skills and implement the behavior specified by the user. This process follows the same call-and-return strategy, and recomposes the skills without any further training. It is illustrated in Figure 3, which shows concrete examples of prompts and outputs. More examples are reported in appendix.

## 3.3 MAESTROMOTIF ON NETHACK

We benchmark MaestroMotif on the NetHack Learning Environment (NLE) (Küttler et al., 2020). In addition to being used in previous work on AI feedback, NetHack is a prime domain to study hierarchical methods, due to the fact that it is a long-horizon and complex open-ended system, containing a rich diversity of situations and entities, and requiring a vast array of strategies which need to be combined for success. To instantiate our method, we mostly follow the setup of Motif (Klissarov et al., 2024), with some improvements and extensions. We now describe the main choices for instantiating MaestroMotif on NetHack, reporting additional details in Appendix A.

**Skills definition** Playing the role of agent designers, we choose and describe the following skills: the `Discoverer`, the `Descender`, the `Ascender`, the `Merchant` and the `Worshipper`. The `Discoverer` is tasked to explore each dungeon level, collect items and survive any encounters. The `Descender` and `Ascender` are tasked to explore and specifically find staircases to either go up, or down, a dungeon level. The `Merchant` and the `Worshipper` are instructed to find specific entities in NetHack and interact with them depending on the context. These entities are shopkeepers for the `Merchant`, such that it attempts to complete transactions, and altars for the `Worshipper`, where it may identify whether items are cursed or not. The motivation behind some of these skills (for example the `Descender` and `Ascender` pair) can be traced back to classic concepts such as bottleneck options (Iba, 1989; McGovern & Barto, 2001; Stolle & Precup, 2002).

**Datasets and LLM choice** To generate a dataset of preferences $\mathcal{D}_{\omega_i}$ for each one of the skills, we mostly reproduce the protocol of Klissarov et al. (2024), and independently annotate pairs of observations collected by a Motif baseline. Additionally, we use the Dungeons and Data dataset of unannotated human gameplays (Hambro et al., 2022b). We use Llama 3.1 70B (Dubey et al., 2024) via vLLM (Kwon et al., 2023) as the LLM annotator, prompting it with the same basic mechanism employed in Klissarov et al. (2024).

**Annotation process** In the instantiation of Motif presented in Klissarov et al. (2024), preferences are elicited from an LLM by considering a single piece of information provided by NetHack, the `messages`. Although this was successful in deriving an intrinsic reward that was generally helpful to play NetHack, our initial experiments revealed that this information alone does not provide enough context to obtain a set of rewards that encode more specific preferences for each skill. For this reason, we additionally include some of the player's `statistics` (i.e., dungeon level and experience level), as contained in the observations, when querying the LLM. Moreover, we leverage the idea proposed by Piterbarg et al. (2023a) of taking the difference between the current state and a state previously seen in the trajectory, providing the difference between states 100 time steps apart as the representation to the LLM. This provides a compressed history (i.e. a non-Markovian representation) to LLM and reward functions, while preventing excessively long contexts.

**Coding environment and Policy Over Skills** A fundamental component of MaestroMotif is an LLM coder that generates Python code (Van Rossum & Drake Jr, 1995). MaestroMotif uses Llama 3.1 405b to generate code that is executed in the Python interpreter to yield initiation and termination functions for the skills, the train-time policy over skills, and the policies over skills employed during deployment. In practice, we find it beneficial to rely on an additional in-context code refinement procedure to generate the policies over skills. This procedure uses the LLM to write and run unit tests and verify their results to improve the code defining a policy over skill (see Appendix A.2 for more details). In our implementation, a policy over skills defines a function that returns the index of the skill to be selected. For the training policy, the prompt given to the LLM consists of the list of skills and a high-level description of an exploratory behavior of the type *"alternate between the `Ascender` and the `Descender`; if you see a shopkeeper activate the `Merchant`..."*, effectively transforming minimal domain knowledge to low-level information about a skill's desired state distributions.

**RL algorithm and skill architecture** To train the individual skills, we leverage the standard CDGPT5 baseline based on PPO (Schulman et al., 2017) using the asynchronous implementation of *Sample Factory* (Petrenko et al., 2020). Instead of using a separate neural network for each skill, we train a single network, with the standard architecture implemented by Miffyli (2022), and an additional conditioning from a one-hot vector representing the skill currently being executed. This enables skills to have a shared representation of the environment, while at the same time reducing potential negative effects from a multi-head architecture (see Section 4.3).

## 4 EXPERIMENTS

We perform a detailed evaluation of the abilities of MaestroMotif on the NLE and compare its performance to a variety of baselines. Unlike most existing methods for the NLE, MaestroMotif is a zero-shot method, which produces policies entirely through skill recomposition, without any additional training. We emphasize this in our evaluation, by first comparing MaestroMotif to other methods for behavior specification from language on a suite of hard and composite tasks. Then, we compare the resulting agents with the ones trained for score maximization, and further analyze our method. We report all details related to the experimental setting in Appendix A.5. All results are averaged across

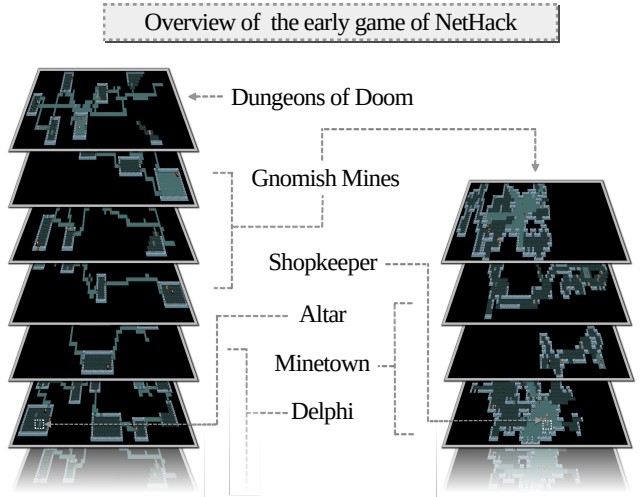

Figure 4: Simplified depiction of the early NetHack game where significant areas (such as branches) and entities are labeled.

nine seeds (for MaestroMotif, three repetitions for skill training and three repetitions for software policy generation), with error bars representing the standard error. All MaestroMotif results are obtained by recombining the skills without training, and the skills themselves were learned only through LLM feedback, without access to other types of reward signals.

**Evaluation suite** As NetHack is a complex open-ended environment (Hughes et al., 2024), it allows for virtually limitless possibilities in terms of task definition and behavior specification. To capture this complexity and evaluate zero-shot control capabilities beyond what has been done in previous work, we define a comprehensive benchmark. We consider a set of relevant, compelling, and complex tasks related to the early part of the game, deeply grounded in both the original NLE paper (Küttler et al., 2020) and the broader NetHack community (Moult, 2022). Our benchmark includes three types of tasks: *navigation* tasks, asking an agent to reach specific locations in the game; *interaction* tasks, asking an agent to interact with specific entities in the game; *composite* tasks, asking the agent to reach sequences of goals related to its location in the game and game status. In navigation and composite tasks, we evaluate methods according to their success rate; in interaction tasks, we evaluate methods according to the number of collected objects. Figure 4 presents an overall depiction of navigation and interaction tasks, and Appendix A.6 provides more details. Our evaluation is also inspired by SkillHack (Matthews et al., 2022), a benchmark for evaluating skill transfer on NetHack.

### 4.1 PERFORMANCE EVALUATION

**Baselines** We measure the performance of MaestroMotif on the evaluation suite described above. For MaestroMotif to generate a policy, it is sufficient for a user to specify a task description in natural language. For this reason, we mainly compare MaestroMotif to methods that are instructable via language: first, to using Llama as a policy via ReAct (Yao et al., 2022), which is an alternative zero-shot method; second, to methods that require task-specific training via RL, with reward functions

|  | Zero-shot | | Task-specific training | | Reward Information |
|---|---|---|---|---|---|
| Task | MaestroMotif | LLM Policy | Motif | Emb. Simil. | RL w/ task reward + score |
| Gnomish Mines | **46% ± 1.70%** | 0.1% ± 0.03% | 9% ± 2.30% | 3% ± 0.10% | 3.20% ± 0.27% |
| Delphi | **29% ± 1.20%** | 0% ± 0.00% | 2% ± 0.70% | 0% ± 0.00% | 0.00% ± 0.00% |
| Minetown | **7.2% ± 0.50%** | 0% ± 0.00% | 0% ± 0.00% | 0% ± 0.00% | 0.00% ± 0.00% |
| Transactions | **0.66 ± 0.01** | 0.00 ± 0.00 | 0.08 ± 0.00 | 0.00 ± 0.00 | 0.01% ± 0.00% |
| Price Identified | **0.47 ± 0.01** | 0.00 ± 0.00 | 0.02 ± 0.00 | 0.00 ± 0.00 | 0.00% ± 0.00% |
| BUC Identified | **1.60 ± 0.01** | 0.00 ± 0.00 | 0.05 ± 0.00 | 0.00 ± 0.00 | 0.00% ± 0.00% |

Table 1: Results on navigation tasks and interaction tasks. MaestroMotif and LLM policy are zero-shot methods requiring no data collection or training on specific tasks; task-specific training methods generate rewards from text specifications (based on AI feedback or embedding similarity) and train an agent with RL; the last column reports the performance of a PPO agent using privileged reward information, a combination of the task reward and the game score (not accessible to the other methods). MaestroMotif largely outperforms all baselines, which struggle with complex tasks.

generated by using either AI feedback or cosine similarity according to the embedding provided by a pretrained text encoder (Fan et al., 2022). In addition, we also compare to an agent trained to maximize a combination of the task reward and the game score (as auxiliary objective), which has thus access to privileged reward information compared to the other approaches. For all non-zero-shot methods, training runs of several GPU-days are required for each task before obtaining a policy.

**Results on navigation and interaction tasks** Table 1 shows that MaestroMotif outperforms all the baselines, which struggle to achieve good performance, in navigation and interaction tasks. Notice that this happens despite the disadvantage to which MaestroMotif is subject when compared to methods that are specifically trained for each task. The poor performance of the LLM Policy confirms the trend observed by previous work (Klissarov et al., 2024): even if the LLM has enough knowledge and processing abilities to give sensible AI feedback, that does not mean that it can directly deal with low-level control and produce a sensible policy via just prompting. At the same time, methods that automatically construct a single reward function that captures a language specification break apart for complex tasks, resulting in a difficult learning problem for an agent trained with RL. MaestroMotif, instead, still leverages the ability of LLMs to automatically design reward functions, but uses code to decompose complex behaviors into sub-behaviors individually learnable via RL.

**Results on composite tasks** A feature of language is its compositionality. Since, in the type of system we consider, a user specifies tasks in language, different behavior specifications can be composed. For instance, an agent can be asked to first achieve a goal, then another one, then a last one. The *composite* category in our benchmark captures this type of task specifications. In Table 2, we compare MaestroMotif to other promptable baselines, showing the task description provided to the methods and their success rate. MaestroMotif has lifted the problem of solving a task to the

| Tasks / Methods | **Golden Exit** *"Alternate between the first three levels of the Dungeons of Doom (at least once) until you collect a minimum of 20 gold pieces and defeat 25 monsters; finally try to quit NetHack"* | **Level Up & Sell** *"Do not leave the first dungeon level until you achieve XP level 4, then find a shopkeeper and sell an item that you have collected; finally survive for another 300 steps."* | **Discovery Hunger** *"Reach the oracle level (the Delphi) in the Dungeons of Doom, but not before discovering the Gnomish Mines and eating some food there after getting hungry."* |
|---|---|---|---|
| MaestroMotif | **24.80 % ± 1.18 %** | **7.09% ± 0.99%** | **7.91% ± 1.47%** |
| LLM Policy | 0% ± 0.00% | 0% ± 0.00% | 0% ± 0.00% |
| Motif | 0% ± 0.00% | 0% ± 0.00% | 0% ± 0.00% |
| Embedding Similarity | 0% ± 0.00% | 0% ± 0.00% | 0% ± 0.00% |

Table 2: Description of the composite tasks and success rate of MaestroMotif and baselines. Using a code policy allows MaestroMotif to compose skills by applying sophisticated logic, requiring memory or reasoning over a higher-level time abstraction. This is impossible to achieve for a zero-shot LLM policy, and hard to learn via a single reward function, which explains the failures of the baselines.

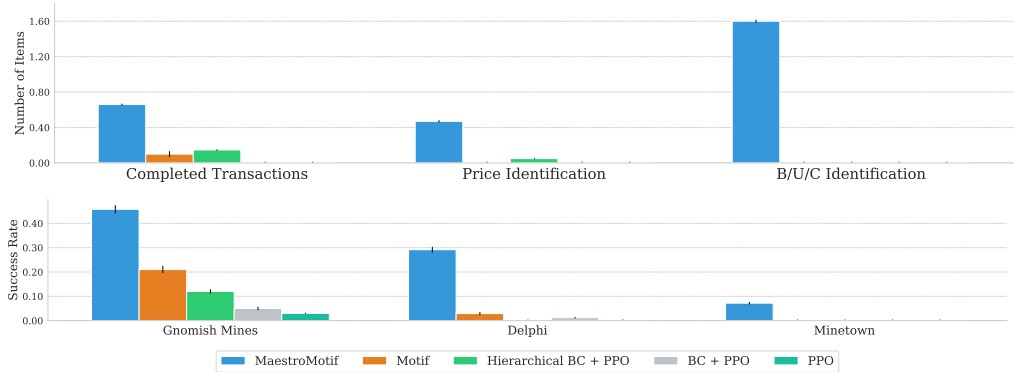

Figure 5: Performance of MaestroMotif and score-maximizing baselines in interaction tasks (first row) and navigation tasks (second row). Despite collecting significant amounts of score, score-maximizing approaches only rarely exhibit any interesting behavior possible in our benchmarking suite.

one of generating a code policy: thus, even if the tasks entail extremely long-term dependencies, simple policies handling only a few variables can often solve them. In contrast, defining rewards that both specify complex tasks and are easily optimizable by RL is extremely hard for existing methods, because exploration and credit assignment in such a complex task become insurmountable challenges for a single low-level policy. To the best of our knowledge, MaestroMotif is the first approach to be competitive at decision-making tasks of this level of complexity, while simultaneously learning to interact through the lowest-level action space. Figure 6 reports an example of complex behavior exhibited by an agent created by MaestroMotif while solving one of the tasks. The overall results, aggregated over navigation, interaction and composite tasks, are presented in Figure 1.

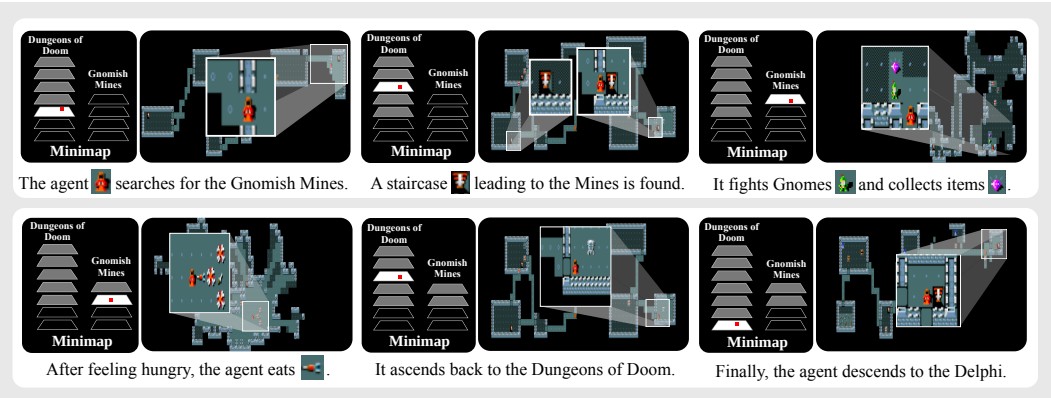

Figure 6: Illustration of MaestroMotif on the composite task `Hunger Discovery`. We show screenshots from the game as well as an accompanying Minimap, where the agent's position is shown as a red dot ■. To complete the task, the agent needs to find the entrance to the Gnomish Mines, which is a staircase randomly generated anywhere between the levels 2 to 4 in the main branch, the Dungeons of Doom. After exploring the first few levels, the agent finally finds the hidden entrance and descends into the Mines, where it fights monsters and collects items to help it survive. After a few hundred turns, the agent's hunger level increases to `hungry`, prompting it to eat a comestible item. Finally, it has to ascend back into the main branch, before beginning the perilous journey down to the Delphi, which appears anywhere, randomly, between the level 5 to 9 in the Dungeons of Doom.

## 4.2 COMPARISON TO SCORE MAXIMIZATION

The vast majority of previous work on the NetHack Learning Environment has focused on agents trained to maximize the score of the game (Sypetkowski & Sypetkowski, 2021; Piterbarg et al., 2023b; Wolczyk et al., 2024). Although the score might seem like a potentially rich evaluation signal,

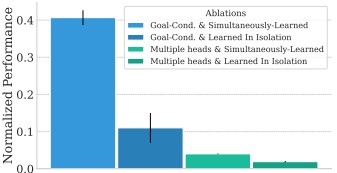 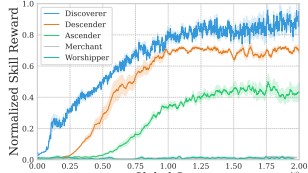 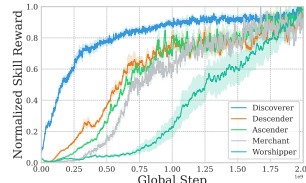

(a) Effect of choices in hierarchical architecture and learning strategy.

(b) Skill reward learning curves (skills learned in isolation).

(c) Skill reward learning curves (skills learned simultaneously).

Figure 8: **(a)** Goal-conditioning as an architecture for skill selection and synchronous alternation of the skills using an exploration policy is essential for obtaining good performance. **(b)** When learning skills asynchronously (alternating them in different episodes), some important skills do not manage to be learned. **(c)** Learning skills synchronously automatically induces an emergent skill curriculum, in which basic skills are learned before the most complex ones.

it has been observed by previous work that a high-performing agent in terms of its score does not necessarily exhibit complex behaviors in the game (Hambro et al., 2022a). To illustrate this fact in the context of our work, we compare MaestroMotif's performance in the navigation and interaction tasks to the one achieved by agents trained to maximize the in-game score via different methods.

Figure 5 reports the performance of these methods, showing that, even if maximizing the score might seem a good objective in NetHack, it does not align to the NetHack community's preferences, even when the source of training signal is an expert, such as in the behavioral cloning case.

## 4.3 ALGORITHM ANALYSIS

Having demonstrated the performance and adaptability of MaestroMotif, we now investigate the impact of different choices on its normalized performance across task categories. Additional experiments can be found in Appendix A.8.

**Scaling behavior** Central to the approach behind MaestroMotif is an LLM producing a policy over skills in code, re-composing a set of skills for zero-shot adaptation. It is known that the code generation abilities of an LLM depend on its scale (Dubey et al., 2024): therefore, one should expect that the quality of the policy over skills generated by the LLM coder will be highly dependent on the scale of the underlying model. We verify this in Figure 7, showing a clear trend of performance improvement for large models. In Appendix A.4, we also investigate the impact of code refinement on the performance of MaestroMotif.

**Hierarchical architecture** As illustrated in Figure 10 of Appendix A.7, the neural network used to execute the skill policies follows almost exactly the same format as the PPO baseline (Miffyli, 2022), with the only difference of an additional conditioning via a one-hot vector representing the skill currently being executed. We found that this architectural choice to be crucial for effectively learning skill policies. In Figure 8a, we compare this choice to representing the skills through different policy heads, as is sometimes done in the literature (Harb et al., 2017; Khetarpal et al., 2020). This alternative approach leads to a collapse in performance. We hypothesize that this effect comes from gradient interference as the different skill policies are activated with different frequencies.

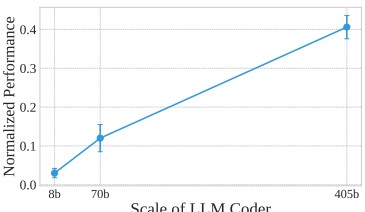

Figure 7: Impact of scaling for the LLM code generator on final performance across tasks.

**Emergent skill curriculum** In Figure 8a, we also verify the importance of learning all the skills simultaneously. We compare this approach to learning each skill in a separate episode. We notice that without the use of the training-time policy over skills, the resulting performance significantly degrades. To better understand the reason behind this, we plot in Figure 8b and Figure 8c, for each skill, the corresponding reward during training. Learning each skill in isolation leads to a majority of the skills not maximizing their own rewards. On the other hand, learning multiple skills in the

same episode leaves space to learn and to leverage simpler skills, opening the possibility of using those simple skills to get to the parts of the environment where it is relevant to use more complex and context-dependent ones, such as the `Merchant` or the `Worshipper`. This constitutes an *emergent skill curriculum*, which is naturally induced by the training-time policy over skills. The curriculum emerges because of the data distribution in which each skill is initiated: a skill expressing a more advanced behavior will only be called by the policy over skills when the appropriate situation can be reached, which will only happen once sufficient mastery of more basic skills is acquired. We discuss in Appendix A.9 how modifying the skill selection strategy, for example by adapting it through online interactions, could further improve the ability to learn skills.

## 5 RELATED WORK

**LLM-based hierarchical control methods**  Our method relates to a line of work which also uses LLMs to coordinate low-level skills in a hierarchical manner. SayCan and Palm-E (Ahn et al., 2022; Driess et al., 2023) also use an LLM to execute unstructured, natural language commands by recomposing low-level skills in a zero-shot manner. A key difference in our work is how the skills are obtained: whereas they leverage a combination of large human teleoperation datasets of language-conditioned behaviors and hand-coded reward functions, we train skills from intrinsic rewards which are automatically synthesized from unstructured observational data and natural language descriptions. MaestroMotif is particularly related to those approaches in which a high-level policy is generated as a piece of code by an LLM (Liang et al., 2023). Voyager (Wang et al., 2024) also uses an LLM to hierarchically create and coordinate skills, but unlike our method, assumes access to control primitives which handle low-level sensorimotor control. LLMs have also been used for planning in PDDL domains (Silver et al., 2023), see Appendix A.10 for a detailed discussion.

**Hierarchical reinforcement learning**  There is a rich literature focusing on the discovery of skills through a variety of approaches, such as empowerment-based methods (Klyubin et al., 2008; Gregor et al., 2017), spectral methods (Machado et al., 2017; Klissarov & Machado, 2023) and feudal approaches (Dayan & Hinton, 1993; Vezhnevets et al., 2017) Most of these methods are based on learning a representation, which is then exploited by an algorithm for skill learning (Machado et al., 2023). In MaestroMotif, we instead work in the convenient space of natural language by leveraging LLMs, allowing us to build on key characteristics such as compositionality and interpretability. Interestingly, some of the skills we leverage in our NetHack implementation are directly connected to early ideas on learning skills, such as those based on notions of bottleneck and in-betweeness (Iba, 1989; McGovern & Barto, 2001; Menache et al., 2002; Şimşek & Barto, 2004). Such intuitive notions had not been scaled yet as they are hard to measure in complex environments. This is precisely what MaestroMotif provides: a bridge between abstract concepts and low-level sensorimotor execution.

## 6 DISCUSSION

Modern foundation models possess remarkable natural language understanding and information processing abilities. Thus, even when they are not able to completely carry out a task on their own, they can be effectively integrated into human-AI collaborative systems to bring the smoothness and efficacy of the design of agents to new heights. In this paper, we showed that MaestroMotif is an effective approach for AI-assisted skill design, allowing us to achieve untapped levels of controllability for sequential decision making in the challenging NetHack Environment. MaestroMotif takes advantage of easily provided information (i.e., a limited number of prompts) to simultaneously handle the highest-level planning and the lowest-level sensorimotor control problems, linking them together by leveraging the best of the LLM and the RL worlds.

Like other hierarchical approaches, MaestroMotif is limited in the behaviors it can express by the set of skills it has at its disposal; given a set of skills, a satisfactory policy for a task might not be representable through their composition. Therefore, an agent designer should perform AI-assisted skill design while keeping in mind what behaviors should be eventually expressed by the resulting agents. Despite this inherent limitation, we believe our work provides a first step towards a new class of skill design methods, more effective and with a significantly higher degree of automation than existing ones. More broadly, MaestroMotif also constitutes evidence for the benefits of a paradigm based on human-AI collaboration, which takes advantage of the complementary strengths of both.

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

# A    APPENDIX

## A.1    SKILL REWARDS

We now list and discuss the prompts used for eliciting preferences from the 70b parameters Llama 3.1 model.

---

**Skill reward prompt template**

```
I will present you with two short gameplay descriptions.  First, tell
me about your knowledge of NetHack.  Mention the goal of NetHack.

Write an analysis describing the semantics of each description
strictly using information from the descriptions and your knowledge
of NetHack.  Provide a comparative analysis based on first
principles.

Here is the preference that you should seek: {skill_modifier}.  Above
everything else, categorically refuse to anger or displease your god,
for example by causing them to thunder or boom out.

Finally, respond by explicitly declaring which description best
fits your preference, writing either ("best_description":  1),
("best_description":  2).  If both contain undesirable events, say
("best_description":  None).

{ "description_1":  "{observation 1}" }
{ "description_2":  "{observation 2}" }
```

---

Prompt 1: Prompt template used for eliciting preferences for each skills reward.

For each skill in the set, we use Prompt 1 as basic prompt template, customizing it with different modifiers depending on the skill. This strategy follows very closely the one used in previous work (Klissarov et al., 2024). This prompt utilizes chain-of-thought prompting (Wei et al., 2022): before asking the model to provide any annotations, we encourage it to articulate its understanding of NetHack and describe the game's main goal.

For each of the skills, we modify the {skill_modifier} variable within the template to steer the LLM's preferences towards a distinct behaviour. We present these modifiers in Prompt 2.

To extract labels from the preferences, we search for the LLM's response by using the following regular expression:

```
(?i)\W*best_*\s*description\W*(?:\s*:*\s*)?(?:\w+\s*)?(1|2|none)
```

This expression looks for slight variations of the answer format that we show to the model in the prompt. If the regex fails to produce an answer, we proceed with the conversation using the LLM and employ Prompt 3 to specifically request a response in the desired format. Our overall response rate with the Llama 3.1 model is very high, around 98% for most prompt configurations.

## A.2    POLICY OVER SKILLS

Leveraging the semantic nature of the skill set of MaestroMotif, we use the coding abilities of LLMs to craft a reasonable strategy for their execution. For the high level code policy, we use the largest open source model available, the 405b parameters Llama 3.1 model.

We use the template of Prompt 4 to obtain snippets of code that constitute the high-level policies for different tasks. In it, we present the LLM with the set of skills, a high level definition of each of them and a desired strategy, all in natural language, which the LLM leverages to write its

---

**Prompt skill modifiers**

```
{Discoverer}:  "players that are adventurous but only within the same
dungeon level, for example by fighting monsters, finding gold pieces
or scrolls; but do not drop them.  Categorically refuse going up and
down dungeon levels."

{Descender}:  "the direction of progress is to explore by going down
the dungeon.  It is urgent to do so, strongly avoid staying on the
same level or worse, going higher."

{Ascender}:  "the direction of progress is only by going up a dungeon
level successfully.  Strongly dislike remaining on the same dungeon
level, no matter the consequences."

{Worshipper}:  "strongly encourage players that interact with the
altar, primarily for identifying whether items are cursed or blessed,
rather than for praying to or pleasing their god."

{Merchant}:  "prefer players that negotiate, sell and interact with
shopkeepers.  Be careful not to steal from stores."
```

---

Prompt 2: Skill modifiers used for instantiating variations of Prompt 1.

---

**Retry prompt**

```
So, which one is the best?  Please respond by saying
("best_description":  1), ("best_description":  2), or
("best_description":  None).
```

---

Prompt 3: Prompt provided to the LLM to continue the conversation when the regular expression does not find a valid annotation in the LLM's answer to the original prompt.

`NetHackPlayer` class, representing the high level policy. We employ a form of chain-of-thought prompting to obtain an initial version of this policy defined in code, an example is given in Output 1.

In many situations, this initial attempt does not capture an adequate strategy for skill execution. To avoid such undesirable outcomes, we leverage the LLM's capability for self-refinement through the help of a self-generated unit test. An initial attempt is passed through a unit test, producing a trace of execution as shown in Output 2. We then ask the model whether the produced trace satisfies the strategy. If the answer is yes, the self-refinement procedure stops. If the answer is no, we ask the LLM to reflect on the code it has previously proposed, identify potential flaws in its logic and write an improved version (as shown in Prompt 5). This process is repeated for a maximum of 3 iterations. Such a process is similar to standard refinement prompting strategies for LLMs (Shinn et al., 2023; Madaan et al., 2023).

A key element of the self-refinement strategy is to leverage a unit test that generates traces of execution. This unit test is itself crafted by the LLM through Prompt 6, which presents the LLM with the same list of skills, their description in natural language and a blueprint of the unit test's structure.

This strategy is employed to generate the code exploration policy used to learn the skill policies. This is done by first defining a general `select_skill` method for selecting skills. This method is then leveraged to define the `reach_dungeons_of_doom` and `reach_gnomish_mines` methods which steer the agent between the different branches of NetHack (as shown in 4). We present in Output 3 one of the obtained explorative code policies. In Output 4, we present one the code policies for achieving the `Discovery Hunger` composite task.

---

**Prompt for the train-time policy over skills**

```
You are to write code which defines the method "select_skill" of the
NetHack Player class that selects amongst a set of skills in the
videogame of NetHack.  The set of skills corresponds to {"discoverer",
"descender", "ascender", "merchant", "worshipper"}.

When activated, the Discoverer fully explores the current dungeon,
while fighting off enemies.  The Descender makes its way to a
staircase and goes down.  The Ascender makes its way to a staircase
and goes up.  The Merchant interacts with shopkeepers by selling
its items.  The Worshipper interacts with altars by identifying its
items.

Find a strategy that will let the player explore fully each of the
first few dungeon levels, alternating directions between going all
the way down towards the maximum depth, then going up towards the
first dungeon.  This might get interrupted by the end of the loop or
if the preconditions of worshipper and merchant allow for it.

You can keep track of any other information by assigning values to
other class attributes, but only if that really helps.

Your code will be verified through this unit test.

###
{unit_test}
###

Before writing the code, write a few questions for yourself, and
answer them to make sure you understand the assignment.
In your response make sure to include the code for the class
definition and the unit test in a python markdown.
```

---

Prompt 4: Description of the coding task for the LLM to code the train-time policy over skills.

---

**Retry prompt for code policy**

```
Go through your code, line by line, and reflect on where the issue
arises from.  Use this to modify your code, remove unnecessary parts
or add new elements.
In your response make sure to include the code for the class
definition and the unit test in a python markdown.
```

---

Prompt 5: Prompt used for code self-refinement while producing code policies.

As shown in Prompt 4, the LLM is allowed to create additional attributes to define the code policy over skills. When the LLM defines such attributes, it is afterwards queried to write code for these attributes such that their values are gathered from the NLE.

The average amount of tokens produced for a policy over skills is 9030 tokens according to the Llama 3 tokenizer. Similarly, the average amount of tokens used for the termination and initiation functions is 810 tokens. To query the 405B model, there exists many solutions online and locally, with throughput as high as 969 tokens/second (generating a policy in largely less than a minute even including the refinement process) and cost as low as $3/1M (generating a policy for a few cents).

---

**Initial policy over skills**

```
class NetHackPlayer:
    def __init__(self, max_depth):
        self.max_depth = max_depth
        self.skills = ["discoverer", "descender", "ascender",
                       "merchant", "worshipper"]
        self.direction = 1  # 1 for down, -1 for up

    def merchant_precondition(self):
        # For the purpose of this example, it always returns False
        return False

    def worshipper_precondition(self):
        # For the purpose of this example, it always returns False
        return False

    def select_skill(self, current_skill, dungeon_depth,
                     merchant_precondition, worshipper_precondition):
        if merchant_precondition:
            return'merchant'
        elif worshipper_precondition:
            return 'worshipper'
        elif current_skill == 'discoverer':
            if dungeon_depth < self.max_depth and self.direction == 1:
                return 'descender'
            elif dungeon_depth > 1 and self.direction == -1:
                return 'ascender'
            else:
                self.direction *= -1
                if self.direction == 1:
                    return 'descender'
                else:
                    return 'ascender'
        elif current_skill == 'descender':
            return 'discoverer'
        elif current_skill == 'ascender':
            return 'discoverer'
```

---

Output 1: Example of generated policy over skills before any refinement.

## A.3    INITIATION AND TERMINATION

Finally, we leverage the coding abilities of the LLM to also define the termination and initiation functions of the skills. These quantities, together with the skill policies, define the option tuple from the options framework (see Section 2). The termination function indicates when a skill should finish its execution and the initiation function when it can be selected by the high level policy. As these functions are significantly simpler than the high level policy, we do not leverage the same self-refinement through unit tests. In Prompt 8, we present the prompt used to define the termination function and in Prompt 9 the one to define the initiation function.

## A.4    CODE REFINEMENT

In Figure 9, we further compare the importance of leveraging code refinement through self-generated unit tests. We notice that this leads to improved results when using the 405b LLM, however no significant difference is observed for the smaller models.

---

**Unit test execution trace**

```
Turn 1: Skill = discoverer, Dungeon depth = 1
Turn 2: Skill = descender, Dungeon depth = 2
Turn 3: Skill = discoverer, Dungeon depth = 2
Turn 4: Skill = descender, Dungeon depth = 3
Turn 5: Skill = discoverer, Dungeon depth = 3
Turn 6: Skill = descender, Dungeon depth = 4
Turn 7: Skill = discoverer, Dungeon depth = 4
Turn 8: Skill = descender, Dungeon depth = 5
Turn 9: Skill = discoverer, Dungeon depth = 5
Turn 10: Skill = ascender, Dungeon depth = 4
Turn 11: Skill = discoverer, Dungeon depth = 4
Turn 12: Skill = ascender, Dungeon depth = 3
Turn 13: Skill = discoverer, Dungeon depth = 3
Turn 14: Skill = ascender, Dungeon depth = 2
Turn 15: Skill = discoverer, Dungeon depth = 2
Turn 16: Skill = ascender, Dungeon depth = 1
Turn 17: Skill = discoverer, Dungeon depth = 1
Turn 18: Skill = descender, Dungeon depth = 2
Turn 19: Skill = discoverer, Dungeon depth = 2
Turn 20: Skill = descender, Dungeon depth = 3
```

---

Output 2: Example of output from a unit test written by the LLM.

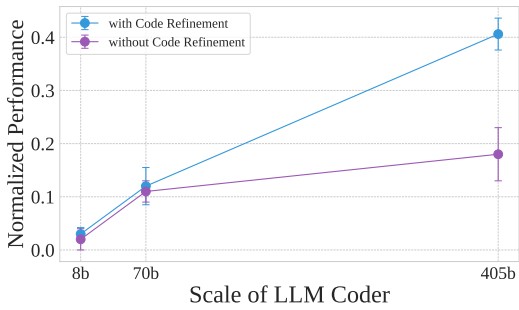

Figure 9: Refining the LLM Coder outputs through a self-generated unit tests yields significant improvements for the 405b parameter Llama model.

## A.5 ENVIRONMENT AND METHOD DETAILS

We base our implementation on the NetHack Learning Environment (Küttler et al., 2020) and Chaotic Dwarven GPT-5 baseline (Miffyli, 2022), which itself was defined on the fast implementation of PPO (Schulman et al., 2017) within Sample Factory (Petrenko et al., 2020). As discussed in Klissarov et al. (2024), although some actions are available to the agent (like the 'eat' action), it is not possible for the agent to actually eat most of the items in the agent's inventory. This limitation is also true for other key actions such as the action for drinking, or the 'quaff' action in NetHack terms. To overcome this limitation, we make a simple modification to the environment by letting the agent eat and quaff any of its items, at random, by performing a particular command (the action associated with the key y). We also include standard actions such as pray, cast and enhance. All agents that we train are evaluated using these same conditions, except the behaviour cloning based agents in Figure 5 which have access to an even larger action set.

For the skill reward training phase of MaestroMotif, we use the message encoder from the Elliptical Bonus baseline (Henaff et al., 2022). Similar to Klissarov et al. (2024), we train the intrinsic reward $r_\phi$ with the following equation,

---

**Unit test prompt**

```
You are to write code for a unit test of the NetHackPlayer class and
its "select_skill" method.  This method takes as input the skill,
"dungeon_depth" and "branch_number" arguments and outputs a skill.
You must write code that simulates how the environment reacts to the
"select_skill" method.

The skills consist of "discoverer", "descender", "ascender",
"merchant", "worshipper".  When activated, the Discoverer fully
explores the current dungeon, while fighting off enemies.  The
Descender makes its way to a staircase and goes down.  The Ascender
makes its way to a staircase and goes up.  The Merchant interacts
with shopkeepers by selling its items.  The Worshipper interacts with
altars by identifying its items.

Here is the template:

" max_depth = 1
player = NetHackPlayer(max_depth)
skill = 'discoverer'
dungeon_depth = 1

for turn in range(20):
print(f"Turn {{turn + 1}}:  Skill = {{skill}}, Dungeon depth =
{{dungeon_depth}}")
merchant_precondition = player.merchant_precondition()
worshipper_precondition = player.worshipper_precondition()
skill = player.select_skill(skill, dungeon_depth,
merchant_precondition, worshipper_precondition)

# the environment updates the dungeon_depth
# Code here

You are to write the unit test only in its current form, not the
NetHackPlayer class.  Do not create new classes, functions or import
anything.
```

Prompt 6: Prompt given to the LLM code generator for coding up the unit test used during refinement.

$$\mathcal{L}(\varphi) = -\mathbb{E}_{(o_1,o_2,y)\sim\mathcal{D}_{\text{pref}}}\Bigg[ \mathbb{1}[y=1]\log P_\varphi[o_1 \succ o_2] + \mathbb{1}[y=2]\log P_\varphi[o_2 \succ o_1]$$
$$+ \mathbb{1}[y=\varnothing]\log\left(\sqrt{P_\varphi[o_1 \succ o_2] \cdot P_\varphi[o_2 \succ o_1]}\right)\Bigg], \quad (2)$$

where $P_\varphi[o_a \succ o_b] = \frac{e^{r_\varphi(o_a)}}{e^{r_\varphi(o_a)}+e^{r_\varphi(o_b)}}$ is the probability of preferring an observation to another. This is the Bradley-Terry model often used in preference-based learning (Thomaz et al., 2006; Knox & Stone, 2009; Christiano et al., 2017). The work on Motif adopted this reward transformation,

$$r_{\text{int}}(\texttt{observation}) = \mathbb{1}[r_\varphi(\texttt{observation}) \geq \epsilon] \cdot r_\varphi(\texttt{observation})/N(\texttt{observation})^\beta, \quad (3)$$

where $N(\texttt{observation})$ was the count of how many times a particular observation has been previously found during the course of an episode. We adopt the same reward transformation, although we relax the requirement that $N()$ is a function over the full course of the episode, but rather over the last 20 steps. This opens the opportunity to leverage this transformation on a larger spectrum

of environments by keeping a short memory of transitions rather than functional forms of counting which are difficult to achieve in many practical settings (Bellemare et al., 2016).

| Hyperparameter | Value |
|---|---|
| Reward Scale | 0.1 |
| Observation Scale | 255 |
| Num. of Workers | 24 |
| Batch Size | 4096 |
| Num. of Environments per Worker | 20 |
| PPO Clip Ratio | 0.1 |
| PPO Clip Value | 1.0 |
| PPO Epochs | 1 |
| Max Grad Norm | 4.0 |
| Value Loss Coeff | 0.5 |
| Exploration Loss | entropy |

Table 2: PPO hyperparameters.

To obtain the LLM-based reward, we train for 20 epochs using a learning rate of $1 \times 10^{-5}$. As Equation 3 shows, we further divide the reward by an episodic count and we only keep values above a certain threshold. The value of the count exponent was 3 whereas for the threshold we used the $85th$ quantile of the empirical reward distribution for each skill, except the Discoverer which used the $95th$ quantile. For the Motif and Embedding Similarity baseline, we perform a similar transformation on their reward, using a count exponent was 3 whereas for the threshold we used the $50th$ quantile. For all methods, before providing the LLM-based reward function to the RL agent, we normalize it by subtracting the mean and dividing by the standard deviation. In the Motif paper, the authors additively combine both the LLM-based intrinsic reward and a reward coming from the environment with a hyperparameter $\alpha$, leading to different trade-offs for different values. In MaestroMotif we completely remove this hyperparameter and instead learn completely through the intrinsic reward coming from the LLM. Finally, in Table 2, we report the remaining standard values of the RL agent's hyperparameters.

### A.6 BENCHMARK DESIGN AND MOTIVATION

We note that out of the original tasks from the NLE paper, the `Staircase` (and closely related `Pet`) tasks have by now been solved (Zhang et al., 2021; Klissarov et al., 2024). The `Score` task is effectively unbounded, but as noted in (Wolczyk et al., 2024), it is possible to achieve very high scores by adopting behaviors which correlate poorly with making progress in the game of NetHack (for example, by staying at early levels and killing weak monsters). This is also an observation corroborated by our experiments in Section 4.2.

To define a set of compelling and useful tasks in the NLE, we take inspiration from the NetHack community, in particular, from the illustrated guide to NetHack Moult (2022). This guide describes various landmarks that every player will likely experience while making progress in the game. Some of these landmarks were also suggested in the original NLE release Küttler et al. (2020). The first such landmark is the `Gnomish Mines` which constitutes the first secondary branch originating in the main branch, the Dungeons of Doom (see Figure 4). The second landmark is `Minetown`, a deeper level into the Gnomish Mines in which players might interact with Shopkeepers and gather items. The third landmark is the `Delphi`, which is a level that appears somewhere between depth 5 and 9 in the main branch and is the home to the Oracle, a famous character in the game. It is not necessary to interact with the Oracle to solve the game of NetHack, but reaching the Delphi is a necessary step towards it, which is the reason we include it and not the Oracle task.

As these tasks are navigation oriented, we additionally include a set of tasks that require the agent to interact with entities found across the dungeons of NetHack. The interactions we select are chosen because they key to the success to any player playing the NetHack game. For this reason, we focus on interactions that will give the agent more information about its inventory of items. In NetHack, most items that are collected have only partially observable characteristics. For example, a ring

that is found could be blessed or cursed, and its magical effects are not revealed (it could be ring of levitation, a ring of cold resistance, etc.).

The first type of interactions are those where the agent interacts with altars associated with the NetHack gods. These offer many benefits, the most common one is the possibility to identify the blessed/cursed/uncursed (B/U/C) status of an item. The difference between a cursed and uncursed item can have deadly consequences in NetHack. The second type of interactions are those where the agent finds a shopkeeper to either sell an item and collect gold, or attempts to sell an item to get an offer from the shopkeeper. When getting a price offer from the shopkeeper, it is possible to identify the kind of item that the agent has in its possession (i.e. a wand of death or a wand of enlightenment).

Overall, we believe that these tasks are well-aligned with making progress towards the goal of NetHack. It is also important to note that even though these tasks are very hard for current AI agents, they only represent a fraction of the complexity of NetHack.

## A.7 HIERARCHICAL ARCHITECTURE

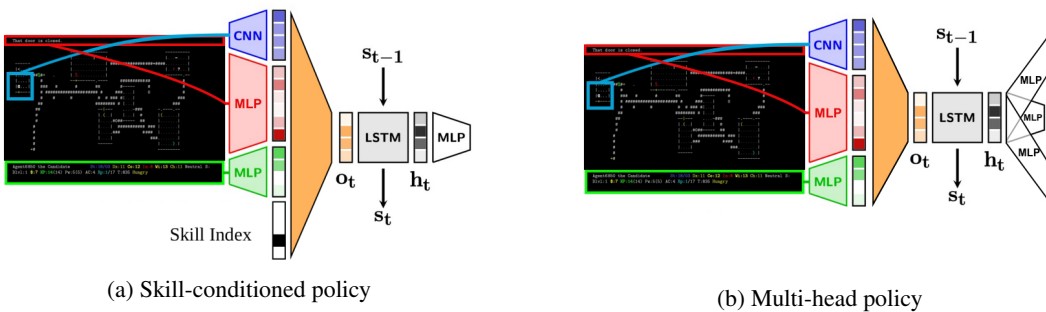

(a) Skill-conditioned policy

(b) Multi-head policy

Figure 10: Neural network architectures. The architecture on left, used throughout the paper, was key for the the successful training of the skill policies.

In Figure 10a we present the architecture used to learn the skill policies, which simply consist of a single neural network conditioned on a one-hot vector. This one-hot vector represents the skill index (i.e. the first entry in this vector is associated with the `Discoverer` skill and the last one with the `Merchant` skill). This implementation is not only efficient in terms of the number of parameters needed to represent a diversity of behaviours, but also was also crucial for successfully learning these behaviours. We explored alternative architectures, such as adding multiple heads to the network, each for one of the skills, as shown in Figure 10b. Results in Figure 8a show that this lead to a collapse in performance which we attribute to a catastrophic interference between the gradients coming from different skills. It is important to notice that the skills are activated with very different frequencies (for example the `Discoverer` is activated almost 50 times more often than the `Worshipper`). Another possibility in terms of architecture would be to consider more sophisticated conditioning mechanism such as FiLM (Perez et al., 2017) which has been successful in various applications.

## A.8 ADDITIONAL ABLATIONS

**Preference elicitation** In Section 3.3, we have presented the ways in which the annotation process used in the NetHack implementation of MaestroMotif differs from the one presented in Klissarov et al. (2024). In Figure 11, we verify how each of these choices affects the final performance of our algorithm. The importance of providing the player statistics within the prompt eliciting preferences from the LLM is made apparent, as without such information the performance drops to almost 30% of its full potential. When the player statistics are provided but no information about how they differ from recent values (i.e. diffStats), the resulting performance is similarly decreased. This is explained by the non-Markovian nature of observations in NetHack: as an example, a status shown as hungry could be the result of being previously satiated or fainting, which present two quite different ways of behaving and would produce difference preferences. Finally, our preference elicitation phase integrates episodes from the Dungeons and Data dataset (Hambro et al., 2022b), which provides greater coverage of possible interactions and observations of NetHack. We notice that this choice is

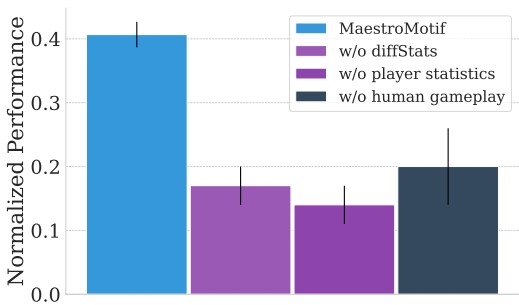

Figure 11: Ablation studies on MaestroMotif's design choices.

important to obtain the full performance of MaestroMotif. This result illustrates how AI feedback can be an effective strategy for leveraging action-free and reward-free datasets.

### A.9 CONSIDERATIONS FOR THE SKILL SELECTION

In this work, we have leveraged an LLM to define a training-time high-level policies, termination and initiation functions in order to learn the skills. These components defining the skills selection strategy were then fixed during the skill learning process. As we have seen in Section 4.3, this led to an emerging curriculum over skills, where easier skills developed first and harder skills developed later on. However, we could see significant improvements in skill learning efficiency if the high-level policies, termination and initiation functions were instead adapted online. This could be done, for example, by deciding what skills to select and how to improve them (Kumar et al., 2024). Ideas from active learning (Daniel et al., 2014; Mendez-Mendez et al., 2023) would be of particular value for pursuing this research direction. Another consideration with respect to the high-level policy is its robustness. Currently, before the high-level policy is deployed, it is verified through a self-generated unit test. This strategy was generally successful to avoid particular failure modes and obtain good strategies. However, it is not a full-proof strategy, and adapting the high-level policy through online interactions could be significantly more robust. One way to approach to adapt the high-level policy would be to provide in-context execution traces from the environment through which the LLM could iterate on a proposed strategy. Another approach would be through RL, for example through intra-option value learning (Sutton et al., 1999). We are then faced with the following question: what reward would this high level policy optimize? A possible answer would be to apply Motif to define such reward function on a per-task basis.

### A.10 ADDITIONAL RELATED WORK

**Connections to the Planning Literature** MaestroMotif learns skills through RL and, when faced with a particular task, re-composes them zero-shot through code that defines the execution strategy. To do so, the LLM writing the code needs to specify where skills can initiate, where they should terminate and how to select between them. MaestroMotif is in fact an instantiation of the options formalism (Sutton et al., 1999; Precup, 2000), which defined the necessary quantities for learning skills in RL. However, the idea to abstract behavior over time in the form of skills has a long history in AI, for example through STRIPS planning (Fikes et al., 1993), macro-operators Iba (1989), Schemas Drescher (1991) and Planning Domain Definition Language (PDDL) (McDermott et al., 1998). The structure behind the option triple can also be seen in related fields, such as formal systems through the Hoare logic (Hoare, 1969). Silver et al. (2023) recently investigate how LLMs can be used as generalized planners by writing programs in PDDL domains, which is similar to how MaestroMotif write code to sequence skills. Their results show that LLMs are particularly strong planners. Another promising direction would be to use LLMs to convert natural language into PDDL, to then leverage classical planning algorithms (Liu et al., 2023). Further investigating the connections between the options framework and symbolic representations would be particularly promising (Konidaris et al., 2018; Bagaria et al., 2021), in particular in the context of LLMs.

| | Zero-shot | | | Task-specific training | |
|---|---|---|---|---|---|
| Task | MaestroMotif | LLM Policy | LLM Policy (Eq. Prompting) | Motif | Motif (Eq. Prompting) |
| Gnomish Mines | **46**% ± **1.70**% | 0.1% ± 0.03% | 0.3% ± 0.03% | 9% ± 2.30% | 9% ± 2.30% |
| Delphi | **29**% ± **1.20**% | 0% ± 0.00% | 0% ± 0.00% | 2% ± 0.70% | 1.7% ± 0.70% |
| Minetown | **7.2**% ± **0.50**% | 0% ± 0.00% | 0% ± 0.00% | 0% ± 0.00% | 0% ± 0.00% |
| Transactions | **0.66** ± **0.01** | 0.00 ± 0.00 | 0.00 ± 0.00 | 0.08 ± 0.00 | 0.09 ± 0.00 |
| Price Identified | **0.47** ± **0.01** | 0.00 ± 0.00 | 0.00 ± 0.00 | 0.02 ± 0.00 | 0.02 ± 0.00 |
| BUC Identified | **1.60** ± **0.01** | 0.00 ± 0.00 | 0.00 ± 0.00 | 0.05 ± 0.00 | 0.04 ± 0.00 |

Table 3: Results on navigation tasks and interaction tasks. We provide all prior knowledge given to MaestroMotif to two additional baselines, LLM Policy (Equivalent Prompting) and Motif (Equivalent Prompting). Results indicate that this additional information does not increase the performance. Learning how and when to leverage this information, from context, makes it very challenging.

**HRL approaches with code policies**  MaestroMotif is particularly related to approaches that combine code to define policies over skills and RL to learn low-level policies, such as *concurrent hierarchical Q-learning* (Marthi et al., 2005), *policy sketches* (Andreas et al., 2017), and *program-guided agents* (Sun et al., 2020). MaestroMotif employs LLMs as generators of reward functions, termination/initiation functions, and policies over skills, significantly simplifying the interaction between humans and the AI system which is used in existing hierarchical RL methods.

## A.11  ADDITIONAL PROMPTING EXPERIMENTS

We further verify the hypothesis that the hierarchical structure of the MaestroMotif algorithm is key to obtain performance. In Table 3, we present two additional baselines. **LLM Policy (equivalent prompting)** based the LLM Policy baseline but its prompt contains all the information that used within the different prompts of MaestroMotif. This includes skill descriptions, high-level descriptions of the task and also the generated code by the policy-over-skills that is used within MaestroMotif. We also investigate **Motif (equivalent prompting)**, which similarly builds on the Motif baseline but provides all the prior knowledge given to MaestroMotif. Despite giving significantly more information to both baselines, the performance does not improve. Although additional information is provided, the burden on how and when to leverage this information, from context, makes it very challenging.

**Code for train-time policy over skills**

```python
class NetHackPlayer:
    def __init__(self, max_depth, branch_depth):
        self.max_depth = max_depth
        self.branch_depth = branch_depth
        self.explored_levels = set()
        self.direction = 'down'  # Start by going down

    def merchant_precondition(self):
        # Placeholder for actual merchant precondition logic
        return False

    def worshipper_precondition(self):
        # Placeholder for actual worshipper precondition logic
        return False

    def select_skill(self, current_skill, dungeon_depth,
                     merchant_precondition, worshipper_precondition):
        if merchant_precondition:
            return 'merchant'
        if worshipper_precondition:
            return 'worshipper'

        if current_skill == 'discoverer':
            self.explored_levels.add(dungeon_depth)
            if self.direction == 'down':
                if dungeon_depth < self.max_depth:
                    return 'descender'
                else:
                    self.direction = 'up'
                    return 'ascender'
            elif self.direction == 'up':
                if dungeon_depth > 1:
                    return 'ascender'
                else:
                    self.direction = 'down'
                    return 'descender'
        elif current_skill == 'descender':
            return 'discoverer'
        elif current_skill == 'ascender':
            return 'discoverer'
        else:
            return 'discoverer'

    def select_skill_dungeons_doom(self, current_skill, dungeon_depth,
            branch_number, merchant_precondition, worshipper_precondition):
        if dungeon_depth == self.branch_depth:
            if branch_number == 2:
                return 'ascender'
            else:
                return 'descender'
        elif branch_number == 2 and dungeon_depth == self.branch_depth + 1:
            return 'ascender'
        else:
            return self.select_skill(current_skill, dungeon_depth,
                    merchant_precondition, worshipper_precondition)

    def select_skill_gnomish_mines(self, current_skill, dungeon_depth,
            branch_number, merchant_precondition, worshipper_precondition):
        if branch_number == 0:
            if dungeon_depth == self.branch_depth:
                return 'descender'
            elif dungeon_depth == self.branch_depth + 1:
                return 'ascender'
        elif branch_number == 2:
            return self.select_skill(current_skill, dungeon_depth,
                merchant_precondition, worshipper_precondition)
        return self.select_skill(current_skill, dungeon_depth,
                merchant_precondition, worshipper_precondition)
```

Output 3: Code of the train-time policy over skills generated by MaestroMotif.

---

**Prompt for the policy over skills for `Discovery Hunger`**

You are to write code which defines the method "select_skill" of the
NetHack Player class that selects amongst a set of skills in the
videogame of NetHack. The set of skills corresponds to {"discoverer",
"descender", "ascender", "merchant", "worshipper"}.

When activated, the Discoverer fully explores the current dungeon,
while fighting off enemies. The Descender makes its way to a
staircase and goes down. The Ascender makes its way to a staircase
and goes up. The Merchant interacts with shopkeepers by selling
its items. The Worshipper interacts with altars by identifying its
items.

Here is your current implementation of the NetHackPlayer class:

```
###
{current_class_implementation}
###
```

You are faced with the task following task.

Reach the oracle level (the Delphi) in the Dungeons of Doom, but not
before discovering the Gnomish Mines and eating some food there after
getting hungry.

The Dungeons of Doom (branch number 0) constitute the main branch
starting from depth 1. At branch_depth, a second staircase appears
opening the door towards the Gnomish Mines (branch number 2) at
branch_depth+1. Therefore, the Gnomish Mines constitute a second
branch from branch_depth+1.

You should make use of the two existing methods. The
"reach_gnomish_mines" method brings the player to the Gnomish Mines
from the Dungeons of Doom. The "reach_dungeons_of_doom" method brings
the player to the Dungeons of Doom from the Gnomish Mines.

Finding food happens by activating the Discoverer skill in the
Gnomish Mines. Immediately after finding the food, head as fast as
possible for the Oracle level in the Dungeons of Doom (branch number
0), without more exploration.

Your code will be verified through this unit test.

```
###
{unit_test}
###
```

Before writing the code, write a few questions for yourself, and
answer them to make sure you understand the assignment.

Identify if you need to define any additional attributes. If you
define additional attributes, for each of them you must write methods
that will simulate changes to their values. These methods must
be called upon in the unit test (not in perform_task), as if the
environment is updating them. The values should also be printed at
each iteration.

In your response make sure to include the code for the class
definition and the unit test in a python markdown.

---

Prompt 7: Description of the coding task for the LLM to code the policy over skills at deployment time when attempting to solve `Discovery Hunger`.

**Code policy for `Discovery Hunger`**

```python
class NetHackPlayer:
    def __init__(self, max_depth, branch_depth):
        self.max_depth = max_depth
        self.branch_depth = branch_depth
        self.skills = ["discoverer", "descender", "ascender", "merchant", "worshipper"]
        self.direction = 1  # 1 for down, -1 for up
        self.eaten_food = False
        self.explored_gnomish_mines = False
[...]
    def select_skill(self, current_skill, dungeon_depth,
            merchant_precondition, worshipper_precondition):
        if merchant_precondition:
            return'merchant'
        elif worshipper_precondition:
            return 'worshipper'
        elif current_skill == 'discoverer':
            if dungeon_depth < self.max_depth and self.direction == 1:
                return 'descender'
            elif dungeon_depth > 1 and self.direction == -1:
                return 'ascender'
            else:
                self.direction *= -1
                if self.direction == 1:
                    return 'descender'
                else:
                    return 'ascender'
        elif current_skill == 'descender':
            return 'discoverer'
        elif current_skill == 'ascender':
            return 'discoverer'

    def reach_gnomish_mines(self, current_skill, dungeon_depth, branch_number,
            merchant_precondition, worshipper_precondition):
        if dungeon_depth == self.branch_depth:
            return 'descender'
        elif dungeon_depth == self.branch_depth + 1 and branch_number == 0:
            return 'ascender'
        else:
            return self.select_skill(current_skill, dungeon_depth,
                    merchant_precondition, worshipper_precondition)

    def reach_dungeons_of_doom(self, current_skill, dungeon_depth, branch_number,
            merchant_precondition, worshipper_precondition):
        if dungeon_depth == self.branch_depth and branch_number == 0:
            return 'descender'
        elif dungeon_depth == self.branch_depth + 1 and branch_number == 2:
            return 'ascender'
        elif dungeon_depth > self.branch_depth and branch_number == 0:
            return self.select_skill(current_skill, dungeon_depth,
                    merchant_precondition, worshipper_precondition)
        else:
            return self.select_skill(current_skill, dungeon_depth,
                    merchant_precondition, worshipper_precondition)

    def perform_task(self, current_skill, dungeon_depth, branch_number,
            merchant_precondition, worshipper_precondition):
        if not self.explored_gnomish_mines:
            if branch_number == 2:
                self.explored_gnomish_mines = True
                return 'discoverer'
            else:
                return self.reach_gnomish_mines(current_skill, dungeon_depth,
                        branch_number, merchant_precondition, worshipper_precondition)
        elif not self.eaten_food:
            self.eaten_food = True
            return 'discoverer'
        else:
            if branch_number!= 0:
                return self.reach_dungeons_of_doom(current_skill, dungeon_depth,
                        branch_number, merchant_precondition, worshipper_precondition)
            elif dungeon_depth < 9:
                return 'descender'
            else:
                return 'discoverer'
```

Output 4: Example of code generated by MaestroMotif to solve the `Discovery Hunger` composite task.

---

**Termination function prompt**

```
You are to implement the skill_termination method of the NetHackPlayer
class.  This method determines when any of the skills should
terminate.

Here is a description of the skills.  When activated, the Discoverer
fully explores the current dungeon, while fighting off enemies.  The
Descender makes its way to a staircase and goes down.  The Ascender
makes its way to a staircase and goes up.  The Merchant interacts
with shopkeepers by selling its items.  The Worshipper interacts
with altars by identifying its items.  If the any of preconditions of
the Merchant or Worshipper in the preconditions is true, the current
skill should terminate no matter what.

def skill_termination(self, skill, skill_time, current_depth,
previous_depth, preconditions)
```

---

Prompt 8: Prompt given to the LLM code generator for the generation of the termination function for each skill.

---

**Initiation function prompt**

```
You are to implement the precondition method of the NetHackPlayer
class.  This method determines when any of the skills can initiate.

Here is a description of the skills.  When activated, the Discoverer
fully explores the current dungeon, while fighting off enemies.  The
Descender makes its way to a staircase and goes down.  The Ascender
makes its way to a staircase and goes up.  The Merchant interacts
with shopkeepers by selling its items.  The Worshipper interacts with
altars by identifying its items.  Define the preconditions only for
the last two skills.  Before writing the code, identify the entities
that will be useful to identify:  mention their ascii characters and
their ascii encoding number.  To correctly identify an entity, you
also have to make use of the the char_ascii_colors that represents the
color of the ascii character.  Refer to color_map to fetch the right
color.

def skill_precondition(self, char_ascii_encodings, char_ascii_colors,
num_items, color_map):
# char_ascii_encodings :  a numpy array representing the ascii
encoding of the characters surrounding the player
# char_ascii_colors :  a numpy array representing the colors of the
characters surrounding the player
# num_items :  the number of items the agents has
# color_map :  a map from common characters to their color
```

---

Prompt 9: Prompt given to the LLM code generator for the generation of the initiation function for each skill.

