# OpenReview forum: "MaestroMotif: Skill Design from Artificial Intelligence Feedback"
_ICLR.cc/2025/Conference — ICLR 2025 Oral_

### Official Review · Reviewer_BKw5 · 2024-10-23

**Soundness:** 4
**Presentation:** 4
**Contribution:** 3
**Rating:** 8
**Confidence:** 4

**Summary:**

The paper presents MaestroMotif, a novel method for AI-assisted skill design in RL that enables zero-shot adaptation to log-horizon tasks in NetHack. The key idea is using LLMs to map human-generated description of skills and high level policies to corresponding to skill specific reward functions, initiation and termination functions on the first hand, and high-level code-based policies on the other hand.
The method is evaluated on NetHack, where it significantly outperforms existing approaches on a series of tasks.

Overall I really like the approach and I think this paper provides an interesting contribution to the field. This said, there are several missing information and several points I'd like to see discussed in the paper (see weaknesses and questions fields).

I'm going to give a rating of 6 for now and I'm willing to increase it if the authors address my concerns below.

Update: I increased the score to an 8 after the rebuttal (see discussion below)

**Strengths:**

The paper is well written and the approach is well motivated.

The approach allows to construct high-performing policies for complicated, long-horizon tasks with high-level human inputs, which is quite impressive.

Many connections with the option framework, makes the paper easy to parse and understand.

I liked the experiments comparing different methods for skill learning and the emphasis on transfer learning across different skills learned in the same network.

Methods are rather clear and the appendix contains a lot of important information.

**Weaknesses:**

There is not much discussion about how robust the system is to the quality of human feedback, and my guess is that the system could be quite brittle.

The reliance on pre-collected expert / exploratory trajectories to generate the skill-specific reward function is a limitation of the approach that is not discussed.

Results are limited to a single environment, and the authors do not discuss how this approach would work in other environments, where it might be useful or not, what the cost of generating this human feedback is in the first place, etc.

The authors did not specify whether the code will be released publicly.

**Questions:**

These are questions and comments

Experiments comparisons:
* MaestroMotif is first training the skills, which costs GPU time too. So we should ideally compare this training resources and sample efficiency to the one of training a policy for the final downstream task. For fair comparisons, the "RL from scratch" approaches should be given the same training time and environment interactions as is used for MaestroMotif pretraining; is that the case?
* I do get the argument that when targeting a new task, there is no additional cost for MaestroMotif and there is one for the "RL from scratch" approach, but this also assumes that the test task can be decomposed with the existing skills, which is conveniently the case here, but might not be in general.
* Is the performance reported for task-specific training the training performance (with exploration noise), or performance measured outside of training with no exploration noise? In some environments where one mistake can be costly, using training performance with exploration noise would under-estimate true performance (eg crossing a bridge requires walking straight but exploration noise could throw you off).

Online skill learning
* The approach here assumes access to a list of skill descriptions that when combined, allow to achieve any testing task, this is mentioned in the discussion.
* This is an important assumption and I would like to see a discussion of how to move past it.
  * Can we imagine the LLM generating candidate skill descriptions instead of using human-generated ones, and learn these? How good would that be? Could that only work if conditioned on a set of test tasks, or could an LLM intuitively figure out a good set of skills that can easily be combined to solve many downstream tasks?
  * What happens if a test task requires skills that were not pre-trained? The current architecture cannot recover from that, but could we imagine letting the LLM generate a missing skill description, pre-training a new skill and then adapting to that OOD task? Would that work?
* Providing feedback about how to best execute the skills in the training phase seems rather unnatural for a human user, could we imagine a way of automatic this process?
* Maybe we could learn a high-level transition model: which termination state distribution leads to which initiation state distribution, ie a graph model of skill chaining. Once we have that we could generate paths in that graph optimizing for some intrinsic metrics (active learning)? This paper does something a bit in that vein: https://arxiv.org/pdf/2402.15025

Data annotation to synthesize reward functions:
* "we independently annotate pairs of observations collected by a Motif baseline" → this sentence sounds like the authors did some of the annotation, is that the case?
* The fact that we need a dataset of expert / exploratory trajectories to train the reward functions so we can explore better seems like a limitation of this approach that should be discussed. How could the reward functions be synthesized without such data?
* How many trajectories are required to enable the training of the reward functions? What are the requirements in terms of data quality / diversity?

Robustness of the approach:
* The high-level policy is generated zero-shot (with couple of refinements), but what happens if it fails? Could the authors discuss how one could train this high-level policy? Maybe the algorithm sometimes needs to go back to the pretraining phase and add a new skill or refine a skill, maybe it needs to try another high-level strategy, how could it detect what's wrong and improve?
* There is no discussion about the impact of human-generated feedback on the success of the approach. How much time and effort has been put in the design of this feedback? Would it work with feedback generated by non-AI researchers NetHack experts? Here the skill descriptions seem rather finetuned.

Results:
* Why is the Reward Information baseline performing worse than the Motif task-specific training? Doesn't it have access to the true task reward? What are the reward functions Motif generates here and how do these explain the higher performance?

Links with the planning literature
* The paper makes interesting connections with the option framework but doesn't mention connections with the planning literature, which I feel might be just as relevant. I especially think of PDDL approaches and task and motion planning (hierarchical planning)
* MaestroMotif indeed resembles approaches using LLM to generate PDDL domain descriptions: where skills are operators with initial and terminal functions, and the LLM is used to generate the high-level policy (bypassing planning) conditioned on code-based conditioned (equivalent to PDDL predicates), eg https://arxiv.org/pdf/2305.11014
* Chaining issues: although this does not seem to occur in the experiments, pretrained skills can often have the problem where the distribution of final state of an operator (/option), is different from the initiation states of the skill we may want to chain it with, such that the next skills stochastically can or cannot be applied. How would your approach handle this? Would it require coming back to the definition of termination function to have them be more constrained?

Typo: implemenetation (L508)

---

> ### Author Response · Authors · 2024-11-21
>
> Thank you for your feedback!
>
> >There is not much discussion about how robust the system is to the quality of human feedback, and my guess is that the system could be quite brittle.
>
> We believe there might be in a misunderstanding. **The annotation effort is done directly by an LLM**, in our case Llama 3.1 70b, and not by a human. (i.e., we use AI feedback, not human feedback). The task of converting from a language description of the skills to actual policies is therefore **entirely carried out automatically, without any human annotations.** We believe this is a strong argument in favour of our method, which makes learning skills significantly more scalable than previous approaches.
>
> To better communicate this, we will modify the following sentence at L237 to clearly specify that the annotation is done from an LLM: "we mostly reproduce the protocol of (Klissarov et al., 2024), and **use an LLM** independently annotate pairs of observations collected by a Motif baseline." The annotation task itself in fact requires no human effort.
>
> >The reliance on pre-collected expert / exploratory trajectories to generate the skill-specific reward function is a limitation of the approach that is not discussed.
>
> Please see our response below, where we answer in detail this point and its associated questions.
>
> >Results are limited to a single environment, and the authors do not discuss how this approach would work in other environments, where it might be useful or not
>
> We presented the general methodology in section 3.1 and 3.2, and separated this from the specific choices made within NetHack. We believe (as highlighted by reviewer QXfj) that this methodology is fundamentally generalizable. We decided to focus on a specific domain as previous work did, to go deeper into the analyses of the behavior of our method.
>
> At its core, MaestroMotif builds on Motif and on the capabilities of LLMs to write code. There already exists extensions of Motif in different domains, such as robotics ([RL-VLM-F](https://rlvlmf2024.github.io)) and web agents ([DigiRL](https://digirl-agent.github.io)), which future work could build on to apply MaestroMotif to a variety of domains. We are also witnessing weekly releases of improved coding abilities in LLMs, which further increases the possibility to apply MaestroMotif elsewhere.
>
>
> >The authors did not specify whether the code will be released publicly.
>
> We will release all the material that can help reproducibility and future research. This includes all the code used for MaestroMotif in order to reproduce our experiments. We will also release the trained skill neural network, the learned reward functions and the LLM annotations/dataset.
>
> >MaestroMotif is first training the skills, which costs GPU time too. So we should ideally compare this training resources and sample efficiency to the one of training a policy for the final downstream task. For fair comparisons, the "RL from scratch" approaches should be given the same training time and environment interactions as is used for MaestroMotif pretraining; is that the case?
>
> This is indeed the case, each of the task-specific methods (Motif, Embedding Similarity and RL w/ task reward + score) are given 4B timesteps during which they learn the task, which is exactly the same amount used by MaestroMotif to learn the skills. Notice that these baselines are given this budget for each of the tasks, whereas MaestroMotif, as the reviewer points, only uses it once.
>
> > I do get the argument that when targeting a new task, there is no additional cost for MaestroMotif and there is one for the "RL from scratch" approach, but this also assumes that the test task can be decomposed with the existing skills, which is conveniently the case here, but might not be in general.
>
> As inherent to any hierarchical approach (e.g., SayCan) and stated in our paper, it is indeed true that a given set of skills will allow the agent to solve a specific family of tasks. Accordingly, we are not making the argument that a finite set of skills will solve any test tasks. However, we believe that MaestroMotif allows a way more efficient way of learning a new set of skills if the agent designer wants to tackle a new family of tasks. We think the problem of automatically growing the set of skills is very interesting, albeit challenging, and we are excited about the possibility of addressing it in future work.

---

> > ### Author Response · Authors · 2024-11-21
> >
> > >Is the performance reported for task-specific training the training performance (with exploration noise), or performance measured outside of training with no exploration noise? In some environments where one mistake can be costly, using training performance with exploration noise would under-estimate true performance (eg crossing a bridge requires walking straight but exploration noise could throw you off).
> >
> > For all baselines and the proposed method, the results reported in the paper are calculated by sampling according to the agent's policy distribution over actions and do not add any exploration noise. This policy over actions is learned through the PPO algorithm.
> >
> > >Can we imagine the LLM generating candidate skill descriptions instead of using human-generated ones, and learn these? How good would that be? Could that only work if conditioned on a set of test tasks, or could an LLM intuitively figure out a good set of skills that can easily be combined to solve many downstream tasks?
> >
> > We experimented with this possibility and our preliminary experiments show that, while a set of skills completely designed by an LLM could be viable in restricted settings, current LLMs seem to be maximally effective in systems for AI-assisted Skill Design, in which humans and AI assistant collaborate to create skills.
> >
> > We believe developing systems that are able to propose, validate, and refine skills is an important and exciting direction for future work. MaestroMotif could provide a stepping stone towards that direction, since it automated some essential component of a hierarchical pipeline for agent design.
> >
> > >What happens if a test task requires skills that were not pre-trained? The current architecture cannot recover from that, but could we imagine letting the LLM generate a missing skill description, pre-training a new skill and then adapting to that OOD task? Would that work?
> >
> > This is a valid and interesting proposal. Indeed, our current approach assumes that the task distribution is known a priori, and that an agent designer can describe a set of skills that could help in solving tasks from that distribution. We hope that future systems, that would incorporate many crucial elements from MaestroMotif, will be based on an iteration loop that manages a library of skills, by proposing and validating them from data.
> >
> >
> > >Providing feedback about how to best execute the skills in the training phase seems rather unnatural for a human user, could we imagine a way of automatic this process?
> >
> > We believe there might be in a misunderstanding. **The annotation effort is done directly by an LLM**, in our case Llama 3.1 70b, not by a human. We will modify the following sentence at L237 to clearly specify that the annotation is done from an LLM: "we mostly reproduce the protocol of (Klissarov et al., 2024), and **use an LLM** independently annotate pairs of observations collected by a Motif baseline." The annotation task itself in fact requires **no human effort.**
> >
> > >Maybe we could learn a high-level transition model: which termination state distribution leads to which initiation state distribution, ie a graph model of skill chaining. Once we have that we could generate paths in that graph optimizing for some intrinsic metrics (active learning)? This paper does something a bit in that vein: https://arxiv.org/pdf/2402.15025
> >
> > The literature of skills chaining [1, 2, 3] has many ramifications that would be interesting to try for  automatically learning the skills. It would be quite interesting to investigate the synergies between LLM-based skill learning and such tabula rasa methods. We are glad to add a paragraph about these connections and cite the mentioned paper.
> >
> > >"we independently annotate pairs of observations collected by a Motif baseline" → this sentence sounds like the authors did some of the annotation, is that the case?
> >
> > Apologies for the confusion, this is not the case. The annotation is done entirely through the LLM, the 70B model.

---

> > > ### Author Response · Authors · 2024-11-21
> > >
> > > >The fact that we need a dataset of expert / exploratory trajectories to train the reward functions so we can explore better seems like a limitation of this approach that should be discussed. How could the reward functions be synthesized without such data? \[...\] How many trajectories are required to enable the training of the reward functions? What are the requirements in terms of data quality / diversity?
> > >
> > > The version of MaestroMotif we presented in our paper indeed relies on a dataset of observations coming from either humans or trained agents. However, we think this is not fundamentally limiting for our approach, for the following three main reasons:
> > > 1. The needed data is purely observational data, with no actions or external rewards. In other words, it is data that can be seen as in-the-wild videos in the domain of interest. While coverage is indeed helpful, this does not need to be data coming from experts. We believe that, for many domains, scraping or collecting such freeform data can be relatively easy.
> > > 2. To reiterate on how expensive expertise or diversity are not needed for our approach to work, a useful reference are the results presented in Figure 14 of the original Motif paper. These results show that the basic technique we employ for learning individual skills is remarkably robust to relatively low levels of "expertise" (as quantified by maximum game score) and diversity (as quantified by the variance in the score distribution of the trajectories in the dataset). Therefore, Motif is robust to datasets that are far from perfect; MaestroMotif, that uses Motif as an internal routine, naturally inherits this robustness.
> > > 3. Recent work \[1\] has shown that, with tiny engineering adjustments, it is possible for Motif to learn reward functions alongside a policy, by using the data collected by an agent online and without the need of any form of external data collected before running the algorithm. The same approach can be generalized to MaestroMotif, that uses Motif as an internal routine, and would amount to using the dataset cumulatively collected by training the skills to simultaneously train the set of corresponding reward functions. We hope future work can explore this combination to further automate the skill design task.
> > >
> > > [1] Zheng, Q., Henaff, M., Zhang, A., Grover, A., & Amos, B. (2024). Online Intrinsic Rewards for Decision Making Agents from Large Language Model Feedback. arXiv preprint arXiv:2410.23022.
> > >
> > > >The high-level policy is generated zero-shot (with couple of refinements), but what happens if it fails? Could the authors discuss how one could train this high-level policy? Maybe the algorithm sometimes needs to go back to the pretraining phase and add a new skill or refine a skill, maybe it needs to try another high-level strategy, how could it detect what's wrong and improve?
> > >
> > > We thank the reviewer for this important question. In our experiments, the high level policy is simple enough that after a few iterations of code refinement the resulting performance is satisfactory. For example, Output 2 in the paper shows how the unit test provides feedback for the LLM to refine the code. This showcases the high level nature of the policy-over-skills, which explains why current LLMs are able to adjust the code correctly.
> > >
> > > That being said, this is not a full-proof solution and we could imagine tasks where failures can still occur. Like the reviewer mentions, there exists many ideas to be explored, such as adding skills and refining them. It is possible one could consider using RL to learn the policy-over-skills, as well as the terminaton/initiation functions. Another possibility would be to provide execution traces from RL agent interacting with the NetHack environment itself. We are happy to include an in-depth discussion around these possibilities for future work.

---

> > > > ### Author Response · Authors · 2024-11-21
> > > >
> > > > >There is no discussion about the impact of human-generated feedback on the success of the approach. How much time and effort has been put in the design of this feedback? Would it work with feedback generated by non-AI researchers NetHack experts? Here the skill descriptions seem rather finetuned.
> > > >
> > > > We remark here that the feedback used to train the reward functions for each skill is purely coming from an LLM, with the human intervention only being a short prompt written to the LLM to perform the annotation task.
> > > >
> > > > For the skill descriptions, our system is explicitly designed for it to be usable by agent designers that are not necessarily AI experts. The information in the prompts is non-technical and could be effectively provided by a domain expert that knows nothing about AI. The amount of time spent on our prompts can be approximately measured to be less than two hours. Indeed, the agent designer simply writes the skill descriptions and the rest is automated: the MaestroMotif algorithm is launched and the resulting performance is returned.
> > > >
> > > > We hope that future research will be able to afford human studies that can better quantify how efficient an approach is at performing AI-assisted Skill Design with humans of diverse levels of domain expertise.
> > > >
> > > >
> > > > >Why is the Reward Information baseline performing worse than the Motif task-specific training? Doesn't it have access to the true task reward? What are the reward functions Motif generates here and how do these explain the higher performance?
> > > >
> > > > We believe one way to understand this difference is to look back at the Motif paper, where the authors show qualitative experiments explaning how Motif help in terms of two fundamental challenges in RL, that is, by helping exploration and credit assignment. The LLM-generated rewards are dense reward functions, just like the true task reward. However, one key difference is that this generated reward is built on prior knowledge contained in an LLM, which can be useful to a larger set of states, for example by predicting future favourable outcomes or by highlighting important aspects of the game.
> > > >
> > > >
> > > > >The paper makes interesting connections with the option framework but doesn't mention connections with the planning literature, which I feel might be just as relevant. I especially think of PDDL approaches and task and motion planning (hierarchical planning)
> > > > MaestroMotif indeed resembles approaches using LLM to generate PDDL domain descriptions: where skills are operators with initial and terminal functions, and the LLM is used to generate the high-level policy (bypassing planning) conditioned on code-based conditioned (equivalent to PDDL predicates), eg https://arxiv.org/pdf/2305.11014
> > > >
> > > > This is a very interesting line of thought. We agree that there are numerous connections between the options framework and the planning literature. In fact, the option triple is a direct descendent of the [Hoare logic](https://en.wikipedia.org/wiki/Hoare_logic#:~:text=Hoare%20logic%20provides%20axioms%20and,Hoare%20and%20many%20other%20researchers.), which is also conceptually aligned with the planning framwork. In fact, one could interpret the LLM-generate code for the high-level policy as a plan, that is later integrated within the algorithm and that produces a policy at execution time. We will add an in-depth discussion to these connections, including the mentioned work, in a revised version of the paper.
> > > >
> > > > >Chaining issues: although this does not seem to occur in the experiments, pretrained skills can often have the problem where the distribution of final state of an operator (/option), is different from the initiation states of the skill we may want to chain it with, such that the next skills stochastically can or cannot be applied. How would your approach handle this? Would it require coming back to the definition of termination function to have them be more constrained?
> > > >
> > > > This is indeed an issue that particularly well illustrated in the Skill Chaining literature. In our implementation of the options framework, we do not require a very strict relation between the termination of one option to the initiation of another. In fact, some of the option's initiation sets are defined over the whole state space, for example the Discoverer. This choice is one way to deal with the issue of chaining: always have at least one option be available. Another direction would be, as the reviewer mentions, to constrain further the termination function. There is important work that could be done to understand the trade-offs of these two very different approaches to chaining.
> > > >
> > > > **Other minor corrections**: Thanks for the feedback! We will fix these editorial issues in the new version of the paper.

---

> > > > > ### Comment · Reviewer_BKw5 · 2024-11-22
> > > > >
> > > > > Thank you for these thorough answers. I'm happy to hear the code will be released.
> > > > >
> > > > > What are requirements on the quality of skill descriptions, how much time did this take to make it work? Can we ask non-expert humans to generate these descriptions and would it work? Or does it require expert knowledge of how LLMs work and how MotifMastro works? The fact that LLMs cannot generate set of skills that work may be the sign that skill descriptions need to be formatted in very specific way for this whole approach to work, which undermines the claim that any human can use the system. When the authors say "For the skill descriptions, our system is explicitly designed for it to be usable by agent designers that are not necessarily AI experts," I see no experiment showing that this is the case, as I assume current skill descriptions have been generated by the authors themselves (expert in the MasteroMotif codebase) and I'm assuming there has been significant iteration on these descriptions too. A human study would have been a strong argument here.
> > > > >
> > > > > I appreciate the discussion on the features the dataset of trajectories should have to make it work, but still believe sufficient coverage and quantity is required. It's remains deep learning so you'd need enough positive and negative examples for each skill, which requires to already be able to achieve these skills. That's also why I believe the online setup may be tricky. The paper pointed to suggests that this can be done online, and that would be great to see it work for MaestroMotif too.
> > > > >
> > > > > I'm generally satisfied by the answers provided by the authors. I believe this approach is interesting and performs well. As we've seen in the above discussions, there is much room left for improvement and extensions of the proposed approach, especially:
> > > > > * A human-study demonstrating ease-of-use
> > > > > * A solution for the online training of skills and the online adaptation of the skill set
> > > > > * Improving the learning of the high-level policy
> > > > >
> > > > > This said, I believe that the work as is will inspire others to extend the approach, which is a good thing.
> > > > > To reflect these point I'll raise my score to an 8 after I see the updated version of the paper.

---

> > > > > > ### Author Response · Authors · 2024-11-24
> > > > > >
> > > > > > We thank the reviewer for their thoughtful comments and questions. We agree that the points the reviewer has raised are particularly important. While addressing them comprehensively within the scope of this submission is challenging, but we are excited about the possible future work that would study them.
> > > > > >
> > > > > > As agreed to, we have updated the paper with respect to the elements we have discussed: all new modifications to the text are highlighted in blue. As the main text is already at the 10-page limit, we were only able to include the detailed discussions in the appendix, with references present in the main text (any significantly longer comments would get us to 11 pages). If the camera ready version allows for an extended format, we will include these discussions within the main text directly.

---

### Official Review · Reviewer_QXfj · 2024-10-27

**Soundness:** 3
**Presentation:** 4
**Contribution:** 4
**Rating:** 10
**Confidence:** 4

**Summary:**

This work presents a system for learning skills that humans describe in language. The system, called MaestroMotif, (1) trains per-skill reward functions using an LLM annotator of interaction datasets, (2) generates code describing the initiation and termination functions of skills, (3) generates code describing how to choose between skills during "training time", and (4) trains a skill-conditioned RL policy based on PPO. There are multiple experiments on the complex NetHack environment, where the proposed system is significantly stronger than other baselines.

**Strengths:**

- The experiments are comprehensive, covering most baselines one would expect and extensive ablations (like comparing skills learned in isolation vs simultaneously, LLM scale, and the architecture). Furthermore, the results clearly demonstrate the effectiveness of MaestroMotif.
- To me, the results of this paper mark a big improvement in the domain of hierarchical RL, where LLM-generated code is used to conduct high-level planning while a small RL network controls low-level actions.
- The proposed technique is quite general, being applicable across many domains outside of NetHack.
- The writing is very clear and the presentation is excellent overall.

**Weaknesses:**

- The technique only trains reward functions on an existing offline dataset, which may be out of distribution with respect to the distribution of states encountered by the trained RL agent. This choice is borrowed from the original Motif paper (which said that the reward model was not fine-tuned during RL training due to simplicity), but based on the poor performance of the RL baselines (used to collect the original dataset) it seems like there may be a greater distribution shift in this work. Results determining whether the trained reward model generalizes to the online distribution would be appreciated (i.e. checking how much the annotator agrees with the reward model on trajectories generated by the final policy).
- The technique relies heavily on the LLM's ability to generate code. However, even with unit tests and code refinement this seems to be difficult for many LLMs according to Figure 9. It is unclear from the paper whether the 405B model has achieved close to "perfect" performance or if there are still key issues with the generated code. Further analysis of the failure modes of the LLM coders would be useful for readers.

**Questions:**

- MaestroMotif performs very well relative to all baselines, but still performs poorly in settings like Minetown, where only a 7.2% success rate is achieved. What are the bottlenecks to better performance? I.e. is the RL policy unable to complete the skills in this setting or is the generated "skill chooser" code bad for some tasks?
- What architecture is used for the reward functions? Prior RLHF literature initialized RMs from smaller LLMs, but it would be interesting to see if different sized RMs would lead to different results from RL training.

---

> ### Author Response · Authors · 2024-11-21
>
> Thank you for your feedback!
>
> >The proposed technique is quite general, being applicable across many domains outside of NetHack.
>
> We are grateful that the reviewer sees the generality of the proposed method; we certainly hope to see future work extending such approaches to robotics and web agents.
>
> >The technique only trains reward functions on an existing offline dataset, which may be out of distribution with respect to the distribution of states encountered by the trained RL agent. This choice is borrowed from the original Motif paper (which said that the reward model was not fine-tuned during RL training due to simplicity), but based on the poor performance of the RL baselines (used to collect the original dataset) it seems like there may be a greater distribution shift in this work. Results determining whether the trained reward model generalizes to the online distribution would be appreciated (i.e. checking how much the annotator agrees with the reward model on trajectories generated by the final policy).
>
> Note that, despite we are using an offline dataset, we take advantage of a highly diverse dataset of human data (please see **Datasets and LLM choice** in Section 3.3), which reduces the chances that the reward functions will get inputs that are completely out of distribution. Indeed, we believe that, if that was the case, even the skills trained by MaestroMotif, and not only the Motif baselines, would be hard to learn. The fact that MaestroMotif can reliably use the reward functions learned from the annotated dataset to train its skills is a strong sign of the adequacy of the data.
>
> >The technique relies heavily on the LLM's ability to generate code. However, even with unit tests and code refinement this seems to be difficult for many LLMs according to Figure 9. It is unclear from the paper whether the 405B model has achieved close to "perfect" performance or if there are still key issues with the generated code. Further analysis of the failure modes of the LLM coders would be useful for readers.
>
> The performance of the 405B model is indeed not perfect, a quantitative indication being that exists a variation of performance, albeit small, across the seeds (we repeat the experiment from scratch a few times). When looking deeper into the generated code, we notice that it produces sometimes slightly different behaviour. Overall, these discrepancies are small enough such that, in the end, the skills are still learned successfully. We will include such code generations and discuss them in the paper.
>
> >MaestroMotif performs very well relative to all baselines, but still performs poorly in settings like Minetown, where only a 7.2% success rate is achieved. What are the bottlenecks to better performance? I.e. is the RL policy unable to complete the skills in this setting or is the generated "skill chooser" code bad for some tasks?
>
> We believe the main reason why the sucess rate is bottlenecked mostly by the ability of the skill policies face a variety of situations in the game. NetHack is this incredibly complex domain that is procedurally generated, partially observable, and stochastic. For example, reaching Minetown first requires finding the staircase leading to the Gnomish Mines, which can be generated anywhere between levels 2 and 4. This would mean that the agent has to alternate between these levels and cover as much ground in each of them in order to find this specific staircase. During all this time, difficult monsters are likely to be generated (since the agent is no more in the safety of level 1), traps have to be avoided and hidden hallways have to be uncovered. To make further significant progress, we believe that a multimodal LLM, one that could see and understand such difficulties (which are not always apparent in the messages and the player statistics), would be key to obtain even more precise reward models to train more capable policies.
>
> >What architecture is used for the reward functions? Prior RLHF literature initialized RMs from smaller LLMs, but it would be interesting to see if different sized RMs would lead to different results from RL training.
>
> The architecture for the reward model is the same one used in Motif, which is based on the [encoder](https://github.com/facebookresearch/motif/blob/51b5f47312ef4cecab65ac3f0df384bc76fae555/rl_baseline/encoders_nle.py#L150) from the Torchbeast baseline (a relatively small neural network). We initially explored the idea using a larger reward model for more precise control (specifically we used both a BERT encoder or a Llama encoder) but we did not see a significant difference in performance at the time.

---

> > ### Comment · Reviewer_QXfj · 2024-11-23
> >
> > Thank you for the response! I think this work is a very strong follow-up to Motif due to its new hierarchical learning results. There are still many clear future directions, like discovering what skills need to be learned and using multimodal models, but I think this paper is very well scoped and shows the clear advantage of decomposing the learning process into high-level LLM generated code and low-level RL policies.

---

> > > ### Author Response · Authors · 2024-11-24
> > >
> > > We thank once again the reviewer for their time and effort, and are appreciative of them seeing the value of this submission!

---

### Official Review · Reviewer_ygS6 · 2024-10-28

**Soundness:** 3
**Presentation:** 3
**Contribution:** 2
**Rating:** 8
**Confidence:** 4

**Summary:**

The paper introduces MaestroMotif, an extension of the recent Motif framework that accepts datasets of unlabeled demonstrations to first learn a set of labeled options, then query an LLM to write Python programs that trigger these learned options to satisfy an arbitrary objective in the given domain. MaestroMotif is compared to Motif, ordinary PPO, and expert behavior-cloning to see how well each method can satisfy arbitrary unseen objectives, and the results indicate that MaestroMotif is better than these baselines at solving long-horizon, complex requests (e.g., "go to area X, obtain item Y, wait until condition Z is met and then take action A").

The authors also present an ablation of MaestroMotif for different approaches to learning options/skills for the framework, showing the importance of learning their skills jointly and without separate policy heads. The authors also look at the scaling performance of MaestroMotif, showing that smaller LLMs produce predictably smaller returns on the complex tasks.

I thank the authors for their thorough rebuttal and for their comments on the work. After reviewing the rebuttal, I have updated my score to 8 and advocate for acceptance of the work.

**Strengths:**

* MaestroMotif fits in well with the presented prior work and literature, and is a logical extension of much of the "code-as-policies" literature from LLMs, combining human demonstrations and data with AI planning and code-synthesis.
* The baselines used (LLM as policy, PPO, etc.) are sensible baselines and suitable cover the different methods that one might employ to attempt to learn arbitrary NetHack policies for zero-shot transfer to new skills or tasks.
* The paper presents an approach that combines large labeled datasets, human expertise, and LLM code synthesis/planning into one framework, neatly exemplifying human-AI collaboration for task completion with LLMs.
* The full details for reproducibility are in the appendix (importance of unit tests, specific prompts, re-prompting strategies, etc.), which is a useful resource for future work.

**Weaknesses:**

* Learning the options networks seems to require intense manual effort, as a human must label different states with preferences/task-alignment to learn different options.
* While MaestroMotif outperforms the baselines for arbitrary tasks, it does seem to be pitted against methods that have no real way of generalizing to such tasks (for example, are the score-maximization approaches capable of receiving sub-goals or goals in any way?). The success of MaestroMotif is noteworthy, but the delta over prior work is less impressive given the prior work is, at times, apparently severely hamstrung.
* The overall framework relies on significant prompting/re-prompting of a 405b model, which could quickly become prohibitively expensive. It would be nice to see results or metrics on the cost (in time, memory, or money) of running MaestroMotif for each of the specified tasks, or for generalization to a new task or domain.
* Much of this paper is tailored to NetHack, so the line between "core requirement/contribution of MaestroMotif" and "tweak made to adapt to NetHack" is blurry.

**Questions:**

* Are the baselines in any way capable of accepting goals, or is their failure to generalize to new tasks fairly expected behavior?
* What is the cost of running MaestroMotif for a single task (i.e., how much time and memory is required, or what type of hardware is needed?)
* How would MaestroMotif be applied to a new domain? My interpretation is: 1) identify necessary core skills (ask an expert), 2) gather a large unlabeled dataset of gameplay, 3) label that dataset with preferences for each task, 4) learn options over the data, 5) prompt an LLM to combine the learned options. Have I missed something?

---

> ### Author Response · Authors · 2024-11-21
>
> Thank you for your feedback!
>
> > Learning the options networks seems to require intense manual effort, as a human must label different states with preferences/task-alignment to learn different options.
>
> We believe there might be in a misunderstanding. **The annotation effort is done directly by an LLM**, in our case Llama 3.1 70b, and not by a human. (i.e., we use AI feedback, not human feedback). The task of converting from a language description of the skills to actual policies is therefore **entirely carried out automatically, without any human annotations.** We believe this is a strong argument in favour of our method, which makes learning skills significantly more scalable than previous approaches.
>
> To better communicate this, we will modify the following sentence at L237 to clearly specify that the annotation is done from an LLM: "we mostly reproduce the protocol of (Klissarov et al., 2024), and **use an LLM** independently annotate pairs of observations collected by a Motif baseline." The annotation task itself in fact requires no human effort.

---

> > ### Author Response · Authors · 2024-11-21
> >
> > > While MaestroMotif outperforms the baselines for arbitrary tasks, it does seem to be pitted against methods that have no real way of generalizing to such tasks (for example, are the score-maximization approaches capable of receiving sub-goals or goals in any way?). The success of MaestroMotif is noteworthy, but the delta over prior work is less impressive given the prior work is, at times, apparently severely hamstrung. \[...\] Are the baselines in any way capable of accepting goals, or is their failure to generalize to new tasks fairly expected behavior?
> >
> > We would like to highlight the results of Table 1, in particular, the ones under "Task-specific training" and "Reward Information". All of these baselines receive human domain knowledge in different ways about the goals to be achieved. For example, the "Motif" baseline leverages the task information within a prompt given to an LLM to annotate pairs of transitions with preferences. These preferences are distilled in a reward function which an RL agent maximizes. The "Embedding Similarity" baseline similarly leverages a natural language description of the task to calculate a cosine similarity between the task description and the current state, which is then used as a reward to guide an RL agent. Finally, the "RL w/ task reward + score" uses the game score as a dense reward and a task specific reward that detects whether the task has been achieved or not. A particular highlight of these experiments is that each of these baselines are trained for 4B timesteps for each of the tasks separately, whereas MaestroMotif, after training the skills, adapts zero-shot to the task. These same baselines are also used within Table 2 for the composite tasks. Additionally, we investigated using an LLM directly as a policy ("LLM Policy" baseline in the paper) to achieve each of the tasks, giving information about the task at hand.
> >
> > To complement the results in the paper, we further explore the LLM Policy and Motif baselines by giving all prior knowledge used within the MaestroMotif algorithm as context to a strong LLM (GPT-4). This prior knowledge include skill descriptions, high-level descriptions of the task and the generated code by the policy-over-skills that is used within MaestroMotif. We refer to this additional baseline as **LLM Policy (Equivalent Prompting)**, which is inspired from related works such as [1]. An additional baseline we provide is to inject this same prior knowledge into the Motif baseline, by prompting the Llama 3.1 model expressing preferences with the skill descriptions, high-level descriptions of the task and the generated code by the policy-over-skills. We refer to this baseline as **Motif (Equivalent Prompting)**. We present the results hereby. We notice that, despite the significantly more information given to both baselines, their performance does not improve. Although additional information is provided, the burden on how and when to leverage this information, from context, makes it very challenging.
> >
> > Overall, we believe that this is a testament to the potential of the options formalism used by MaestroMotif for efficiently leveraging structure in complex environments.
> >
> > | Task    | Zero-shot | Zero-shot | Zero-shot | Task-specific training| Task-specific training |
> > |---|-------|----|------|---|---|
> > |         | MaestroMotif| LLM Policy| LLM Policy (Equivalent Prompting)| Motif | Motif (Equivalent Prompting) |
> > | **Gnomish Mines**       | **46% ± 1.70%**| 0.1% ± 0.03% | 0.3% ± 0.03% | 9% ± 2.30% | 7% ± 2.30% |
> > | **Delphi**              |**29% ± 1.20%**| 0% ± 0.00% | 0% ± 0.00% | 2% ± 0.70% | 1.7% ± 0.70%  |
> > | **Minetown**            | **7.2% ± 0.50%**| 0% ± 0.00% | 0% ± 0.00% | 0% ± 0.00% | 0% ± 0.00% |
> > |         |             |          |            |         |         |
> > | **Transactions**        |**0.66 ± 0.01**| 0.00 ± 0.00 | 0.00 ± 0.00| 0.08 ± 0.00| 0.09 ± 0.00|
> > | **Price Identified**    | **0.47 ± 0.01**| 0.00 ± 0.00 | 0.00 ± 0.00| 0.02 ± 0.00| 0.02 ± 0.00|
> > | **BUC Identified**      | **1.60 ± 0.01**| 0.00 ± 0.00 | 0.00 ± 0.00| 0.05 ± 0.00| 0.04 ± 0.00|
> >
> >
> > | Task    | Zero-shot | Zero-shot | Zero-shot | Task-specific training| Task-specific training |
> > |---|-------|----|------|---|---|
> > |         | MaestroMotif| LLM Policy| LLM Policy (Equivalent Prompting)| Motif | Motif (Equivalent Prompting) |
> > | **Golden Exit** | **24.80% ± 1.18%**| 0% ± 0.00% | 0% ± 0.00% | 0% ± 0.00% | 0% ± 0.00% |
> > | **Level Up & Sell** |**7.09% ± 0.99%**| 0% ± 0.00% | 0% ± 0.00% | 0% ± 0.00% | 0% ± 0.00% |
> > | **Discovery Hunger** | **7.91% ± 1.47%**| 0% ± 0.00% | 0% ± 0.00% | 0% ± 0.00% | 0% ± 0.00% |
> >
> >
> > [1] Nottingham et al., 2024, Skill Set Optimization: Reinforcing Language Model Behavior via Transferable Skills

---

> > > ### Author Response · Authors · 2024-11-21
> > >
> > > > The overall framework relies on significant prompting/re-prompting of a 405b model, which could quickly become prohibitively expensive. It would be nice to see results or metrics on the cost (in time, memory, or money) of running MaestroMotif for each of the specified tasks, or for generalization to a new task or domain. \[...\] What is the cost of running MaestroMotif for a single task (i.e., how much time and memory is required, or what type of hardware is needed?)
> > >
> > >
> > > The average amount of tokens used for a policy over skills is 9030 tokens according to the Llama tokenizer. Similarly, the average amount of tokens used for the termination and initiation functions is 810 tokens. To query the 405b model, there exists many solutions online and locally, with throughput as high as 969 tokens/second (generating a policy in largely less than a minute even including the refinement process; [source](https://cerebras.ai/blog/llama-405b-inference)) and cost as low as $3/1M (generating a policy for a few cents) [source](https://fireworks.ai/pricing). We will include these numbers in the revised version of the paper.
> > >
> > > > Much of this paper is tailored to NetHack, so the line between "core requirement/contribution of MaestroMotif" and "tweak made to adapt to NetHack" is blurry.
> > >
> > >
> > > We presented the general methodology in section 3.1 and 3.2, and separated this from the specific choices made within NetHack. We believe (as highlighted by reviewer QXfj) that this methodology is fundamentally generalizable. We decided to focus on a specific domain as previous work did, to go deeper into the analyses of the behavior of our method.
> > >
> > > At its core, MaestroMotif builds on Motif and on the capabilities of LLMs to write code. There already exists extensions of Motif in different domains, such as robotics ([RL-VLM-F](https://rlvlmf2024.github.io)) and web agents ([DigiRL](https://digirl-agent.github.io)), which future work could build on to apply MaestroMotif to a variety of domains. We are also witnessing weekly releases of improved coding abilities in LLMs, which further increases the possibility to apply MaestroMotif elsewhere.
> > >
> > > > How would MaestroMotif be applied to a new domain? My interpretation is: 1) identify necessary core skills (ask an expert), 2) gather a large unlabeled dataset of gameplay, 3) label that dataset with preferences for each task, 4) learn options over the data, 5) prompt an LLM to combine the learned options. Have I missed something?
> > >
> > > This is indeed a good summary of how MaestroMotif could be applied to a new domain. We believe that our formulation allows a practitioner to better understand where the inputs from an agent designer are needed, compared to other recently proposed approaches exclusively based on prompting.
> > >
> > >
> > > We restate here, to avoid any misunderstanding, that the annotation process in the third step is entirely carried out by an LLM, without any human annotation. This scalable aspect of the method, together with the generality of its core principles (Motif and code generation), make MaestroMotif a particularly promising approach for HRL and decision-making in general.

---

> > > > ### Author Response · Authors · 2024-11-25
> > > >
> > > > Dear Reviewer, we would like to thank you for the time spent on our submission. We are getting close to the end of the discussion phase (ending tomorrow) and wanted to send this gentle reminder. We have done our best to answer the concerns that were raised and we would love to hear back from the reviewer on whether they had the chance to read our responses and whether we have addressed their concerns.

---

### Official Review · Reviewer_KM3c · 2024-11-02

**Soundness:** 2
**Presentation:** 2
**Contribution:** 2
**Rating:** 5
**Confidence:** 2

**Summary:**

The paper introduces a method called MaestroMotif for designing and using AI-assisted skills that enhance agent performance and adaptability, particularly for complex tasks in environments like NetHack. MaestroMotif leverages Large Language Models (LLMs) to translate natural language skill descriptions into reward functions, which are then used to generate code for skill initiation and termination and to create a policy over these skills. This allows the AI to perform specified behaviors through a combination of reinforcement learning (RL) and LLM-based code generation.
The paper evaluates MaestroMotif on the NetHack Learning Environment (NLE) and demonstrates its effectiveness in handling complex navigation, interaction, and composite tasks. MaestroMotif’s zero-shot approach, using skills without additional training, outperformed both LLM-only and reward-based RL baselines, highlighting the value of hierarchical skill composition and the LLM’s role in skill design.

**Strengths:**

Overall, I believe MaestroMotif presents an effective approach to incorporating human prior knowledge to solve specific tasks. Given that NetHack tasks involve a variety of challenges, including understanding natural language descriptions, planning over high-level abstractions, and exploration, it is notable that the authors demonstrated improved performance in the NetHack environment using their method.
Additionally, I find it interesting that the authors showed that learning skills simultaneously is essential for acquiring skills more efficiently, thanks to the emergence of curriculum-based learning.

**Weaknesses:**

- **Source of Performance Gains**

  I wonder whether the performance gain primarily comes from the human domain knowledge or the method itself. It appears that the method relies heavily on human effort and domain knowledge to solve individual tasks. For instance, humans select a set of skills such as *Discoverer, Descender, Ascender, Merchant,* and *Worshipper,* and they even required to modify prompts for acquiring each of these skills. Therefore, when comparing this work with baselines, it’s difficult to determine whether the performance gain is due to the method itself or the expertise of human contributors. While the core of this work aims to incorporate human priors, the baseline algorithms should ideally leverage human priors at a comparable level.

- **Concerns about Generalizability**

  I'm not entirely convinced that the proposed five skills are general enough to tackle any downstream task in the NetHack environment, or if they are instead a highly targeted set of skills specifically suited for the benchmark tasks. To better demonstrate the method’s generalizability, one approach could be to train reusable skills across a broader range of environments, such as robotics tasks. For instance, training foundational skills on a robotic platform, like a humanoid or robotic arm, and performing zero-shot evaluations on new tasks that require combinations of these learned skills could provide stronger evidence of generalizability.

- **Need for Pseudocode**

  It is challenging to understand the entire concept concretely without pseudocode for the full process, and some aspects remain unclear. For example:

    - If certain agent states satisfy instantiation conditions for more than one skill, how does the LLM prioritize among different skills?
    - What happens if none of the skills satisfy the instantiation conditions?
    - Does the training-time policy over skills adapt/evolve during training? I wonder because, as each individual skill is being changed along the training process, it might be helpful for a policy-over-skills adapts to the current set of learned skills.

  I understand that the specific prompts in the Appendix may partially address these questions; however, I believe it would be beneficial to clearly state them in the main sections.

- **Relationship to Related Work**
     - It would be helpful if the relationship between this work and Motif (Kilssarov, 2024) were more explicitly stated.

- **Minor Corrections**
     - *Line 196:* "crafts aploicy" → "crafts a policy"
     - *Line 239:* *Llama 3.1 70b* → *Llama 3.1 70B*
     - *Figure 8(c):* It is difficult to compare the performance of the two different approaches, as they are visualized in separate plots.
     - *Appendix A.2 and A.3* : title of these subsections are same
     - *Line 269:* What negative impact? Additional clarifications would be helpful for better understanding

**Questions:**

Could you please eloborate more about the concerns that I made on the Weakness sections? Especially, follwoing three.

- Source of Performance Gains
- Concerns about Generalizability
- Need for Pseudocode

---

> ### Author Response · Authors · 2024-11-21
>
> Thank you for your feedback!
>
> > I wonder whether the performance gain primarily comes from the human domain knowledge or the method itself.
>
> We would like to highlight the results of Table 1, in particular, the ones under "Task-specific training" and "Reward Information". All of these baselines receive human domain knowledge in different ways about the goals to be achieved. For example, the "Motif" baseline leverages the task information within a prompt given to an LLM to annotate pairs of transitions with preferences. These preferences are distilled in a reward function which an RL agent maximizes. The "Embedding Similarity" baseline similarly leverages a natural language description of the task to calculate a cosine similarity between the task description and the current state, which is then used as a reward to guide an RL agent. Finally, the "RL w/ task reward + score" uses the game score as a dense reward and a task specific reward that detects whether the task has been achieved or not. A particular highlight of these experiments is that each of these baselines are trained for 4B timesteps for each of the tasks separately, whereas MaestroMotif, after training the skills, adapts zero-shot to the task. These same baselines are also used within Table 2 for the composite tasks. Additionally, we investigated using an LLM directly as a policy ("LLM Policy" baseline in the paper) to achieve each of the tasks, giving information about the task at hand.
>
> To complement the results in the paper, we further explore the LLM Policy and Motif baselines by giving all prior knowledge used within the MaestroMotif algorithm as context to a strong LLM (GPT-4). This prior knowledge include skill descriptions, high-level descriptions of the task and the generated code by the policy-over-skills that is used within MaestroMotif. We refer to this additional baseline as **LLM Policy (Equivalent Prompting)**, which is inspired from related works such as [1]. An additional baseline we provide is to inject this same prior knowledge into the Motif baseline, by prompting the Llama 3.1 model expressing preferences with the skill descriptions, high-level descriptions of the task and the generated code by the policy-over-skills. We refer to this baseline as **Motif (Equivalent Prompting)**. We present the results hereby. We notice that, despite the significantly more information given to both baselines, their performance does not improve. Although additional information is provided, the burden on how and when to leverage this information, from context, makes it very challenging.
>
> Overall, we believe that this demonstrates the potential of the options formalism used by MaestroMotif for efficiently leveraging structure in complex environments.
>
> | Task    | Zero-shot | Zero-shot | Zero-shot | Task-specific training| Task-specific training |
> |---|-------|----|------|---|---|
> |         | MaestroMotif| LLM Policy| LLM Policy (Equivalent Prompting)| Motif | Motif (Equivalent Prompting) |
> | **Gnomish Mines**       | **46% ± 1.70%**| 0.1% ± 0.03% | 0.3% ± 0.03% | 9% ± 2.30% | 7% ± 2.30% |
> | **Delphi**              |**29% ± 1.20%**| 0% ± 0.00% | 0% ± 0.00% | 2% ± 0.70% | 1.7% ± 0.70%  |
> | **Minetown**            | **7.2% ± 0.50%**| 0% ± 0.00% | 0% ± 0.00% | 0% ± 0.00% | 0% ± 0.00% |
> |         |             |          |            |         |         |
> | **Transactions**        |**0.66 ± 0.01**| 0.00 ± 0.00 | 0.00 ± 0.00| 0.08 ± 0.00| 0.09 ± 0.00|
> | **Price Identified**    | **0.47 ± 0.01**| 0.00 ± 0.00 | 0.00 ± 0.00| 0.02 ± 0.00| 0.02 ± 0.00|
> | **BUC Identified**      | **1.60 ± 0.01**| 0.00 ± 0.00 | 0.00 ± 0.00| 0.05 ± 0.00| 0.04 ± 0.00|
>
>
> | Task    | Zero-shot | Zero-shot | Zero-shot | Task-specific training| Task-specific training |
> |---|-------|----|------|---|---|
> |         | MaestroMotif| LLM Policy| LLM Policy (Equivalent Prompting)| Motif | Motif (Equivalent Prompting) |
> | **Golden Exit** | **24.80% ± 1.18%**| 0% ± 0.00% | 0% ± 0.00% | 0% ± 0.00% | 0% ± 0.00% |
> | **Level Up & Sell** |**7.09% ± 0.99%**| 0% ± 0.00% | 0% ± 0.00% | 0% ± 0.00% | 0% ± 0.00% |
> | **Discovery Hunger** | **7.91% ± 1.47%**| 0% ± 0.00% | 0% ± 0.00% | 0% ± 0.00% | 0% ± 0.00% |
>
>
> [1] Nottingham et al., 2024, Skill Set Optimization: Reinforcing Language Model Behavior via Transferable Skills
>
> > I'm not entirely convinced that the proposed five skills are general enough to tackle any downstream task in the NetHack environment
>
> As inherent for any hierarchical approach (e.g., SayCan) and stated in our paper, it is indeed true that a given set of skills will allow the agent to solve a specific family of tasks.
>
> However, MaestroMotif allows a significantly more efficient way of learning a new set of skills if the agent designer wants to tackle a new family of tasks. We think the problem of automatically growing the set of skills is very interesting, albeit challenging, and we are excited about the possibility of addressing it in future work.

---

> > ### Author Response · Authors · 2024-11-21
> >
> > > To better demonstrate the method’s generalizability, one approach could be to train reusable skills across a broader range of environments, such as robotics tasks
> >
> > We decided to focus our experiments on a single rich domain, used by the most related work to ours, with the goal of going very deep into the analysis of the method. For example, this allowed us to investigate the zero-shot performance of different approaches on complex and compositional tasks and to have a great number of ablations.
> >
> > At its core, MaestroMotif relies on Motif, which leverages the capability of LLMs to evaluate behaviour. There already exists extensions of Motif to different domains, such as robotics ([RL-VLM-F](https://rlvlmf2024.github.io)) and web agents ([DigiRL](https://digirl-agent.github.io)), which future work could build on to apply MaestroMotif to a variety of domains. We believe, just as reviewer QXfj mentions, that the method is fundamentally generalizable.
> >
> > >It is challenging to understand the entire concept concretely without pseudocode for the full process, and some aspects remain unclear.
> >
> > We do not have a precise pseudocode in our paper because it would require the introduction of more mathematical symbols that would make the presentation of the paper more complex for a reader to parse. We used the diagram in Figure 2 to summarize the various phases of the method, and the prompts in the main paper and the appendix to give more details about how the method works.
> >
> > To further help the reviewer understand the method, we hereby provide an alternative overview of how MaestroMotif performs AI-assisted Skill Design to obtain a set of skills from language descriptions (based on the phases discussed in Section 3 of the paper).
> >
> > -----
> > *Inputs: dataset of interactions, language description of each skill*
> >
> > *Phase 1*:
> > - Train one reward function for each skills through AI feedback annotating the dataset of interactions with the Llama 3.1 70b model for each skill from its description (Prompt 1 in the paper).
> >
> > *Phase 2*:
> > - Use Llama 3.1 405b to write the termination functions for each skill from its description (Prompt 8 in the paper)
> > - Use Llama 3.1 405b to write the initiation functions for each skill from its description (Prompt 9 in the paper)
> >
> > *Phase 3*:
> > - Use Llama 3.1 405b to write the unit test, used to write the policy over skills (Prompt 6 in the paper)
> > - Use Llama 3.1 405b to write the training policy over skills, using the written unit test to iteratively improve this policy (Prompt 4 in the paper)
> >
> > *Phase 4*:
> > - Randomly initialize skill-conditioned neural network (Figure 10a in the paper)
> > - Gather the LLM-based neural reward functions, termination/initiation functions code and policy-over-skills code.
> > - Execute each skill, according to the flow of decisions dictated by the termination/initiation functions and policy-over-skills.
> > - During the execution of each skill, query the neural reward functions to provide feedback to the RL agent.
> > - Update the skill-conditioned neural network with this generated experience using PPO.
> >
> > ----
> >
> > After the review process, we will release all the material that can help reproducibility and future research, including the code to train the skills and reproduce our experiments and the trained skill models.
> >
> > > If certain agent states satisfy instantiation conditions for more than one skill, how does the LLM prioritize among different skills?
> >
> > If multiple skills can be activated from a given state, then the policy over skills produced by the LLM will simply output an index for a skill, according to the control logic implied by the code.
> >
> >
> > > What happens if none of the skills satisfy the instantiation conditions?
> >
> > For our NetHack implementation of MaestroMotif, some of the skills always satisfy the instantiation condition (i.e., the agent can in any moment decide to go to the next or previous level). For any functional HRL method based on the options framework, there should be at least one skill that is available to choose for the policy over skills. When applying MaestroMotif to other domains, one could alternatively choose a default skill that gets activated when no other skill is available.

---

> > > ### Author Response · Authors · 2024-11-21
> > >
> > > > Does the training-time policy over skills adapt/evolve during training? I wonder because, as each individual skill is being changed along the training process, it might be helpful for a policy-over-skills adapts to the current set of learned skills.
> > >
> > > In the current version of MaestroMotif, the policy over skills is fixed for the entire duration of training. Our experiments show that a curriculum for learning skills automatically emerges in this setting, causing more basic skills to be learned before more complex skills (see Figure 8b).
> > >
> > > We believe that a policy over skills that adapts to how the skills are being trained, for instance by de-prioritizing skills whose training is not progressing, can be a very promising direction for future work.
> > > Despite the additional difficulties related to non-stationarity and potential instability, such an adaptive policy over skills could make the learning of the skills more efficient by creating a more effective curriculum.
> > >
> > >
> > > > It would be helpful if the relationship between this work and Motif (Kilssarov, 2024) were more explicitly stated.
> > >
> > > As stated in the paper, "MaestroMotif uses an LLM’s feedback to convert high-level descriptions into skill-specific reward functions, via the recently-proposed Motif approach". In other words, this means that Motif is a method that is used by MaestroMotif (i.e., an internal routine) for automatically creating reward functions for training the skills.
> > >
> > > Different phases of Motif as illustrated in Figure 2 of the Motif paper, map, for each skill, to different phases of MaestroMotif. Phases 1 and 2 of Motif (*dataset annotation* and *reward training*) map to Phase 1 of MaestroMotif, where a different reward function is created for each skill (see panel 1 of Figure 2 in the MaestroMotif paper, depicting *Automated Skills Reward Design*). Phase 3 of Motif (*reinforcement learning training*) maps to the training of each individual skill that is conducted in phase 4 of MaestroMotif (see panel 4 of Figure 2 in the MaestroMotif paper, depicting *Skill Training via Reinforcement Learning*).
> > >
> > > We believe we explicitly refer to the Motif paper throughout our submission. One paragraph where we are glad to add another reference is the one of **RL algorithm and skill architecture** at the end of Section 3.3. We will add a sentence to mention that the RL architecture (CDGPT5) for our experiments is the same as the one in Motif.
> > >
> > > > Line 269: What negative impact? Additional clarifications would be helpful for better understanding
> > >
> > > We will change the paper to directly refer to the ablation in Figure 8a, which directly shows the negative impact of using a skill neural architecture that employs multiple heads.
> > >
> > > **Other minor corrections**: Thanks for the feedback! We will fix these editorial issues in the new version of the paper.

---

> > > > ### Author Response · Authors · 2024-11-25
> > > >
> > > > Dear Reviewer, we would like to thank you for the time spent on our submission. We are getting close to the end of the discussion phase (ending tomorrow) and wanted to send this gentle reminder. We have done our best to answer the concerns that were raised and we would love to hear back from the reviewer on whether they had the chance to read our responses and whether we have addressed their concerns.

---

> ### Comment · Reviewer_KM3c · 2024-11-28
>
> **Dear Authors,**
>
> Thank you for your effort in conducting additional experiments. However, I am afraid that I am still not fully convinced by the claim that the performance gain comes from the proposed method itself, rather than from human priors. In my view, the essence of this work lies in the **decomposition**. The ultimate task requires complex planning, so the proposed method *decomposes* the task into smaller components and trains each skill individually with the assistance of an LLM. Subsequently, a high-level LLM determines how to combine these skills to solve the overarching task.
>
> This raises an important question: **Who determines the appropriate decomposition?** Isn't it the human? According to my understanding, humans select the set of skills—such as *Discoverer, Descender, Ascender, Merchant*, and *Worshipper*. Furthermore, the method requires additional human effort to refine prompts to properly learn each skill. If this decomposition were performed by the LLM rather than by humans, I would agree that the method is scalable and that the performance gain stems from the method itself.
>
> If humans play such a crucial role in the process, I have doubts about the scalability of the method. For instance, let's say we have a different, imaginary video game environment called "XYZ_WONDERLAND." We want to apply MaestroMotif in this new environment. What would be the first step? Humans would need to play the game extensively to master and understand its core mechanics. Only then could they come up with an appropriate decomposition of the ultimate task into manageable tasks. If a framework requires lots of human efforts for individual target task, I regard that as less generalizable.
>
> In the additional experiments, the authors claim to have provided equivalent human priors for all baseline methods. While this may be true in terms of the information provided to the systems, I believe they differ in terms of the **explicitness of incorporating human prior knowledge**. Did the baseline methods also undergo explicit task decomposition, as MaestroMotif did? I suspect they did not. Solving the task as a whole without proper decomposition may be challenging. Thus, decomposition is the key factor, and this is done by humans. A fair comparison would involve decomposing the task into the same five skill sets proposed in the paper, then using different methods (e.g., LLM-policy or Motif) to train each skill and combining them.
>
> This leads me to question: What is the core contribution of this work? If the decomposition is performed by humans, is the contribution limited to learning each skill? But this is already accomplished using existing work such as Motif. Then, is the contribution in using a policy-over-skills approach? However, employing an LLM as a high-level strategy planner has been suggested in many prior works. In this sense, I struggle to identify the core contribution of this paper.
>
> If I have missed anything important, please let me know. I'm really open to adjusting my score if my understanding is incorrect.

---

> > ### Author Response · Authors · 2024-12-01
> >
> > Dear Reviewer, we thank you for your continued discussion during the rebuttal period. We understand your concerns and hereby provide answers. We are afraid there is a misunderstanding about what the contributions of the paper are, and we hope our answer will offer some clarification.
> >
> > > In my view, the essence of this work lies in the decomposition. \[...\] This raises an important question: Who determines the appropriate decomposition? Isn't it the human? \[...\] If humans play such a crucial role in the process, I have doubts about the scalability of the method. For instance, let's say we have a different, imaginary video game environment called "XYZ_WONDERLAND." \[ ... \] This leads me to question: What is the core contribution of this work?
> >
> > The main premise of our paper, as stated in the first sentence of the abstract, is that _"Describing skills in natural language has the potential to provide an accessible way to inject human knowledge about decision-making into an AI system"_. AI-assisted Skill Design, the setting we formally describe in our background section, assumes a human agent designer has enough knowledge about a domain to describe a sensible skill decomposition in natural language.
> >
> > We completely agree that the end goal of this line of research is to eventually have a system that autonomously proposes and learns skills. We have chosen a specific strategy to advance the progress of this field: instead of building a system that tried to address every aspect of the problem at the same time, we build a system that allows us to understand and improve a particular aspect in a controlled way.
> >
> > MaestroMotif does not fully automate yet the process of designing a set of skills, but we believe it is a first step in understanding how the modern AI tools we have at our own disposal can be tied together and improved to automate this process, in *significantly easier way compared to alternatives*. To summarize, we believe our paper to make a substantial contribution because:
> >
> > 1. MaestroMotif is an effective method for AI-assisted Skill Design. There are many situations in which humans have prior high-level knowledge and want to convert it into sequential decision-making abilities; performing AI-assisted Skill Design can be a natural strategy to face the challenges related to those settings.
> > 2. MaestroMotif leverages and extends recent approaches (e.g., Motif, code-generating LLMs), tying them together in harmony to solve part of the problem of automated skill design. Our paper provides in-depth experimental analyses of the properties of such systems. While MaestroMotif solves the problem of training and coordinating the skills and not the one of coming up with high-level ideas for which skills to learn, it makes tangible progress towards the ultimate goal.
> >
> >
> >
> >
> > > A fair comparison would involve decomposing the task into the same five skill sets proposed in the paper, then using different methods (e.g., LLM-policy or Motif) to train each skill and combining them.
> >
> > To address this point, we have ran an additional hierarchical baseline where the skills are instantiated through the LLM Policy method. The set of skills and the selection logic is exactly the same as the one in the MaestroMotif algorithm. We refer to this baseline as the Hierarchical LLM Policy and present results in the following table across all tasks.
> >
> > | Task    | Zero-shot | Zero-shot |
> > |---|-------|-------|
> > |         | MaestroMotif| Hierarchical LLM Policy|
> > | **Gnomish Mines**       | **46% ± 1.70%**| 0.15% ± 0.06% |
> > | **Delphi**              |**29% ± 1.20%**| 0% ± 0.00% |
> > | **Minetown**            | **7.2% ± 0.50%**| 0% ± 0.00% |
> > |         |             |          |
> > | **Transactions**        |**0.66 ± 0.01**| 0.00 ± 0.00 |
> > | **Price Identified**    | **0.47 ± 0.01**| 0.00 ± 0.00 |
> > | **BUC Identified**      | **1.60 ± 0.01**| 0.03 ± 0.01 |
> > |||
> > | **Golden Exit** | **24.80% ± 1.18%**| 0% ± 0.00% |
> > | **Level Up & Sell** |**7.09% ± 0.99%**| 0% ± 0.00% |
> > | **Discovery Hunger** | **7.91% ± 1.47%**| 0% ± 0.00% |
> >
> >
> > The results show a signficant difference in performance between the MaestroMotif method and this baseline, yet these two methods use the same decomposition. This clearly illustrates the contribution of our submission: efficiently learning skills and their execution, given a decomposition. This is also what separates our contribution to existing methods that rely on behavior cloning from large human teleoperation datasets or manually hand-crafted heuristics. MaestroMotif achieves this through key choices, including separating low-level control using RL and high-level planning via in-context learning, as well as crucial architectural and skill learning logic choices, detailed in the Algorithm Analysis of Section 4.3.

---

> > > ### Comment · Reviewer_KM3c · 2024-12-02
> > >
> > > I appreciate your continued efforts throughout the discussion and your additional experiments. While I still have concerns regarding the scalability and generalizability of the method due to its reliance on human effort—and believe that autonomous decomposition of the entire task would significantly enhance the method's generalizability—I now have a clearer understanding of your contribution, thanks to your efforts. I am happy to increase my score.

---

### Author Response · Authors · 2024-11-21

We thank the reviewers for their time and feedback.

We summarize here three important points that we stated in the individual responses.

**Additional baseline**

To complement the results in the paper, we further explore the LLM Policy and Motif baselines by giving all prior knowledge used within the MaestroMotif algorithm as context to a strong LLM (GPT-4). This prior knowledge include skill descriptions, high-level descriptions of the task and the generated code by the policy-over-skills that is used within MaestroMotif. We refer to this additional baseline as **LLM Policy (Equivalent Prompting)**, which is inspired from related works such as [1]. An additional baseline we provide is to inject this same prior knowledge into the Motif baseline, by prompting the Llama 3.1 model expressing preferences with the skill descriptions, high-level descriptions of the task and the generated code by the policy-over-skills. We refer to this baseline as **Motif (Equivalent Prompting)**. We present the results hereby. We notice that, despite the significantly more information given to both baselines, their performance does not improve. Although additional information is provided, the burden on how and when to leverage this information, from context, makes it very challenging.

Overall, we believe demonstrates the potential of the options formalism used by MaestroMotif for efficiently leveraging structure in complex environments.

| Task    | Zero-shot | Zero-shot | Zero-shot | Task-specific training| Task-specific training |
|---|-------|----|------|---|---|
|         | MaestroMotif| LLM Policy| LLM Policy (Equivalent Prompting)| Motif | Motif (Equivalent Prompting) |
| **Gnomish Mines**       | **46% ± 1.70%**| 0.1% ± 0.03% | 0.3% ± 0.03% | 9% ± 2.30% | 7% ± 2.30% |
| **Delphi**              |**29% ± 1.20%**| 0% ± 0.00% | 0% ± 0.00% | 2% ± 0.70% | 1.7% ± 0.70%  |
| **Minetown**            | **7.2% ± 0.50%**| 0% ± 0.00% | 0% ± 0.00% | 0% ± 0.00% | 0% ± 0.00% |
|         |             |          |            |         |         |
| **Transactions**        |**0.66 ± 0.01**| 0.00 ± 0.00 | 0.00 ± 0.00| 0.08 ± 0.00| 0.09 ± 0.00|
| **Price Identified**    | **0.47 ± 0.01**| 0.00 ± 0.00 | 0.00 ± 0.00| 0.02 ± 0.00| 0.02 ± 0.00|
| **BUC Identified**      | **1.60 ± 0.01**| 0.00 ± 0.00 | 0.00 ± 0.00| 0.05 ± 0.00| 0.04 ± 0.00|


| Task    | Zero-shot | Zero-shot | Zero-shot | Task-specific training| Task-specific training |
|---|-------|----|------|---|---|
|         | MaestroMotif| LLM Policy| LLM Policy (Equivalent Prompting)| Motif | Motif (Equivalent Prompting) |
| **Golden Exit** | **24.80% ± 1.18%**| 0% ± 0.00% | 0% ± 0.00% | 0% ± 0.00% | 0% ± 0.00% |
| **Level Up & Sell** |**7.09% ± 0.99%**| 0% ± 0.00% | 0% ± 0.00% | 0% ± 0.00% | 0% ± 0.00% |
| **Discovery Hunger** | **7.91% ± 1.47%**| 0% ± 0.00% | 0% ± 0.00% | 0% ± 0.00% | 0% ± 0.00% |


[1] Nottingham et al., 2024, Skill Set Optimization: Reinforcing Language Model Behavior via Transferable Skills

**Clarity on AI/human feedback**

We noted that, due to some ambiguity in our writing, Reviewers ygS6 and BKw5 might have thought that our reward models are trained using human annotation. We remark here that this is not the case, and that the reward models are entirely trained from annotations coming from an LLM (i.e., via AI feedback), similarly to what was done in the original Motif paper.


**Code release**


We will release all the material that can help reproducibility and future research. This includes all the code used for MaestroMotif in order to reproduce our experiments. We will also release the trained skill neural network, the learned reward functions and the LLM annotations/dataset.

---

### Meta-Review · Area_Chair_g5UA · 2024-12-23

**Metareview:**

The paper presents a method for using textual skill descriptions, together with RL, to learn a hierarchical policy. The paper addresses an important problem and presents a timely and novel approach. The writing and motivation are clear, the evaluation is adequate, and the results show significant benefits. Despite some concerns regarding the method's general applicability given the extensive required domain expertise, the limited scope of the evaluation only on NetHack, and the fairness of baseline comparison, the paper was overall very well received by reviewers.

**Additional Comments On Reviewer Discussion:**

Reviewers and authors discussed the paper thoroughly and mapped out its strengths and limitations.

---

### Decision · Program_Chairs · 2025-01-22

Accept (Oral)